



# What controls the formation of nocturnal low-level stratus clouds over southern West Africa during the monsoon season?

Karmen Babić[1], Norbert Kalthoff[1], Bianca Adler[1], Julian F. Quinting[1], Fabienne Lohou[2], Cheikh Dione[3], and Marie Lothon[2]

[1]Institute of Meteorology and Climate Research, Karlsruhe Institute of Technology (KIT), Karlsruhe, Germany
[2]Laboratoire d'Aérologie, Université de Toulouse, CNRS, UPS, Toulouse, France
[3]African Center for Meteorological Applications for Development, Niamey, Niger

**Correspondence:** Karmen Babić (karmen.babic@kit.edu)

**Abstract.** Nocturnal low-level stratus clouds (LLC) are frequently observed in the atmospheric boundary layer (ABL) over southern West Africa (SWA) during the summer monsoon season. Considering the effect these clouds have on the surface energy and radiation budgets as well as on the diurnal cycle of the ABL, they are undoubtedly important for the regional climate. However, an adequate representation of LLC in the state–of–the–art weather and climate models is still a challenge, which

is largely due to the lack of high-quality observations in this region. In several recent studies, a unique and comprehensive data set collected in summer 2016 during the DACCIWA (Dynamics-Aerosol-Cloud-Chemistry Interactions in West Africa) ground-based field campaign was used for the first observational analyses of the parameters and physical processes relevant for the LLC formation over SWA. However, occasionally stratus-free nights occur during the monsoon season as well. Using observations and ERA5 reanalysis, we investigate differences in the boundary layer conditions during 6 stratus-free and 20

stratus nights observed during the DACCIWA campaign. Our results suggest that the interplay between three major mechanisms is crucial for the formation of LLC during the monsoon season: (i) the onset time and strength of the nocturnal low-level jet (NLLJ), (ii) horizontal cold-air advection and (iii) background moisture level. Namely, weaker or later onset of NLLJ leads to reduced contribution from horizontal cold-air advection. This in turn results in a weaker cooling and thus saturation is not reached. Such deviation in the dynamics of NLLJ is related to the arrival of cold air mass propagating northwards from the

coast called Gulf of Guinea maritime inflow. Additionally, stratus-free nights occur when the intrusions of dry air masses, originating from e.g. central or south Africa, reduce the background moisture over the large parts of SWA. Based on the backward trajectories analysis, another possible reason for clear nights is descending of air originating from drier levels above the marine boundary layer.

## 1 Introduction

The nocturnal boundary layer (NBL) over southern West Africa (SWA) during the summer monsoon season is typically characterized with low-level stratus clouds (LLC), which cover extensive areas stretching over a region of approximately 800 000





km$^2$ at a maximum coverage (van der Linden et al., 2015). Due to the lack of ground-based observations in this region, investigations of LLC characteristics and related processes were mostly performed based on satellite images, synoptic observations and a few modeling studies (Knippertz et al., 2011; Schrage and Fink, 2012; Schuster et al., 2013; van der Linden et al., 2015; Adler et al., 2017). Within the Dynamics-aerosol-chemistry-cloud-interactions over West Africa (DACCIWA) project

(Knippertz et al., 2015), a field campaign was conducted in June and July 2016 (Flamant et al., 2018; Kalthoff et al., 2018). The aim of the DACCIWA ground-based campaign was to collect a comprehensive and high-quality data set (Kalthoff et al., 2018; Bessardon et al., 2018), which enables studies of LLC characteristics, the intranight variability of conditions in the atmospheric boundary layer (ABL) (Kalthoff et al., 2018; Adler et al., 2019; Dione et al., 2018) and physical processes relevant for formation, maintenance and dissolution of LLC (Adler et al., 2019; Babić et al., 2019). Characteristics of the nocturnal

low-level jet (NLLJ), LLC, monsoon flow and Gulf of Guinea maritime inflow, or just maritime inflow for brevity, as well as their interaction during the DACCIWA campaign are analyzed in Dione et al. (2018).

Based on the above mentioned studies, we now have a better understanding of mechanisms and factors which control the formation and maintenance of LLC. The results show that relevant processes include the horizontal advection of cool maritime air from the Gulf of Guinea embedded in the southwesterly monsoon layer, formation of a NLLJ and turbulent mixing related

to the strong wind shear underneath the NLLJ (Schuster et al., 2013; Adler et al., 2017, 2019; Babić et al., 2019). High-resolution simulations and DACCIWA observations suggest that additional processes could be important for LLC formation, which include vertical cold-air advection related to orographically induced lifting on the windward side of the mountains as well as to gravity waves, and enhanced convergence and upward motion upstream of existing clouds (Schuster et al., 2013; Adler et al., 2017, 2019). Cooling is found to be the dominant process for LLC formation, leading to a continuous increase

of relative humidity (RH), until finally saturation is reached and LLC form with a cloud-base height (CBH) near the height of NLLJ maximum (Adler et al., 2017, 2019; Babić et al., 2019). Adler et al. (2019) find that the horizontal cold-air advection contributes about 50 % to the observed cooling prior to the LLC formation, cooling by radiative flux divergence is roughly 20 % and about 22 % by sensible heat flux divergence in the presence of a NLLJ. After the LLC form, turbulent mixing supports the cooling below the cloud base, while strong radiative cooling at the cloud top helps to maintain thick stratus (Schuster et al.,

2013; Babić et al., 2019). Based on the analysis of data gathered during the DACCIWA field campaign, Lohou et al. (2019) developed a conceptual model of LLC formation, maintenance and dissolution over SWA.

However, stratus clouds do not necessarily form every night during the monsoon season (e.g., Kalthoff et al., 2018), even though some of the key features, such as, southwesterly monsoon flow and an embedded NLLJ are present. So far, only a few studies have investigated differences in conditions between nights with and without stratus clouds in West Africa. Schrage

et al. (2007) investigated local and synoptic conditions for 11 cloudy and 12 clear nights using radiosonde measurements from Parakou (Benin, 9°21' N, 2°37' E) and operational model otuput from European Centre for Medium-Range Weather Forecasts (ECMWF). They related cloudy nights to increased overall frictional drag and shear stresses in a neutrally stratified ABL, which remained coupled to the surface. This resulted in a positive net moisture flux convergence due to large-scale wind speed convergence (i.e. changes in the speed of the flow along the streamline) in the ABL. On the other hand, they found clear

nights to occur when a nocturnal inversion caused the decoupling of the boundary layer from the surface. This resulted in a





reduced frictional drag, which coincided with moisture flux divergence north of 6°N. Schrage and Fink (2012) used remote sensing observations between May and October 2006 farther north of Savè at Nangatchori (in Benin, 9.70°N, 1.68°E; 434 m above mean sea level) and they concluded that the shear-generated turbulence beneath the NLLJ axis is the major mechanism explaining stratus formation by reducing the stability of surface layer and causing vertical mixing of moisture. However, open

questions remain since their study did not explain why then the NLLJ is observed on many stratus-free nights, nor why the NLLJ did not form on some non-precipitating cloudy nights. Based on the high-resolution regional simulations performed with the Weather Research and Forecasting model for the July–September 2006 period, Schuster et al. (2013) analyzed differences between 15 cloudiest and 15 clearest nights at 6.2°N. Their model results indicate that cooling during cloudy nights is caused by cold-air advection from the south up to about 1000 m above ground level (a.g.l.), which are also characterized by stronger

vertical mixing having a larger vertical extend as well. They have found that in cloudy nights the layer between 300 and 600 m a.g.l. cools by approximately 1 K more and dries by about 0.3 g kg$^{-1}$ less than for clear nights between 18:00 and 06:00 UTC. They also found a stronger monsoon circulation during cloudy nights.

The analysis of atmospheric conditions and processes during stratus and stratus-free nights has important practical implications with respect to the regional climate, especially considering the challenges that state–of–the–art numerical weather

prediction and climate models encounter in representing extensive and persistent LLC over SWA (Knippertz et al., 2011; Hannak et al., 2017; Kniffka et al., 2019). In this work, we present an observation-based analysis of differences in conditions and processes during stratus and stratus-free nights based on a high-quality, comprehensive data set collected during the DACCIWA campaign. Therefore, our results not only provide the observational verification of previous modeling studies, but also complement the previous observational studies reported farther north of Savè (e.g., Schrage et al., 2007; Schrage and Fink,

2012), thus enabling a comprehensive overview for the whole West Africa. The research questions we address in this study are: (i) What are the main differences in the ABL conditions between nights with and without stratus clouds? (ii) How do the observed differences influence the main processes relevant for the development of LLC? (iii) What are the main factors leading to these differences?

The paper is organized as follows: in Sect. 2 a brief description of the study site, data and methods used is given. In Sect.

3 the spatial distribution of LLC in the DACCIWA region is presented, while in Sect. 4 we compare ABL conditions during stratus and stratus-free nights. Two case studies representing stratus and stratus-free nights, respectively, are presented in Sect. 5, while the large-scale setting and its impact on the formation of LLC is given in Sect. 6. Section 7 provides the discussion of the results, while the main findings are summarized in Sect. 8.

## 2 Data and methods

In this study we analyze the data collected during the DACCIWA ground-based measurement campaign at three supersites in Kumasi (Ghana), Savè (Benin) and Ile-Ife (Nigeria), which took place in June and July 2016. Specifically, we use remote sensing and in-situ data collected at the Savè supersite (Bessardon et al., 2018; Kalthoff et al., 2018), where the "KITcube" mobile platform (Kalthoff et al., 2013) and the instrumentation from the Toulouse University Paul Sabatier (UPS) was installed.





Additionally, we use radiosonde measurements (Maranan and Fink, 2016) performed as part of the DACCIWA radiosonde campaign (Flamant et al., 2018) at three coastal stations, namely in Abidjan (Ivory Coast), Accra (Ghana) and Cotonou (Benin), and two stations inland at Lamto (Ivory Coast) and Parakou (Benin) (Fig. 1). We also use the ERA5 reanalysis data set for the investigation of large-scale conditions over the whole SWA and for the calculation of backward trajectories (Sect. 2.2).

## 2.1 DACCIWA ground-based measurements

The CBH is determined from the attenuated backscatter coefficient profiles measured by the CHM15k ceilometer based on a threshold method (manufacturer Lufft, personal communication, 2016). The CHM15k ceilometer records the backscatter averaged over 60 s up to 15 km a.g.l. and has a vertical resolution of 15 m, while the threshold method can detect up to 3 CBHs and we use only the first one. Figure 2 shows a 30-min frequency distribution of CBHs detected below 500 m a.g.l. measured at Savè during the whole DACCIWA campaign. We note that LLC formed on many nights, however, some stratus-free nights occurred as well. Periods without stratus during the night consist of one night which occurred before the monsoon onset, two consecutive nights at the beginning of post-onset phase and three consecutive nights in July during the vortex phase (Knippertz et al., 2017), resulting in a total of 6 stratus-free nights during the campaign. We note that for typical monsoon conditions (*post-onset phase*, Fig. 2), stratus usually formed during the night. Based on ceilometer measurements, precipitation records and satellite images, the nights with intermittent and scattered LLC are excluded from the analysis, as well as the nights with rain or the presence of mesoscale convective systems at night in the vicinity of the site. In total, 20 nights with LLC and undisturbed conditions are selected for the analysis and comparison with 6 stratus-free nights.

Normal and frequent radiosonde measurements, as well as different continuously running remote sensing instruments, such as ultra-high frequency (UHF) wind profiler (provides the information on the wind speed and direction profiles from 19 June to 30 July 2016) and microwave radiometer (measures potential temperature profiles from 30 June to 30 July 2016) provided high-resolution information of dynamic and thermodynamic conditions in the ABL. Normal radiosondes were launched at Savè every day during the campaign at 05:00 UTC (to correspond to standard synoptic time 06:00 UTC when it reaches the tropopause). Please note that the local standard time for Benin is UTC plus 1 h. During 15 Intensive Observation Periods (IOPs) the radiosondes were launched in regular intervals of 1.5 h, starting at 17:00 UTC prior to the IOP day until 11:00 UTC on the IOP day. At other radiosonde stations normal radiosondes are performed four times daily at standard synoptic times (i.e. at 06:00, 12:00, 18:00 and 00:00 UTC). We also use near-surface measurements at 4 m a.g.l. of standard meteorological parameters, 30-min averaged turbulence fluxes and turbulence variables and radiation fluxes (Kohler et al., 2016).

## 2.2 ERA5 reanalysis data and calculation of backward trajectories

The large-scale conditions during stratus and stratus-free nights affecting the SWA are investigated based on the ERA5 reanalysis data (Copernicus Climate Change, 2017) from the ECMWF with 0.5 degree latitude–longitude grid and 3-hourly temporal resolution. For each stratus and stratus-free night, backward trajectories are calculated using the Lagrangian Analysis Tool (LAGRANTO: Sprenger and Wernli, 2015) for the Savè region (represented by five ERA5 grid points surrounding Savè, i.e., at 8°N/2°E, 8°N/2.5°E, 8°N/3°E, 7.5°N/2.5°E, 8.5°N/2.5°E), which is defined as a 100-km by 100-km box. We used 3-hourly



3-dim wind fields from the ERA5 reanalysis data set. The trajectories are started at 00:00, 03:00 and 06:00 UTC from 10 to 100 hPa above the surface at intervals of 10 hPa and integrated backward for 48 hours. By doing so, the backward trajectories are representative of air masses at the levels in (and immediately above/below) the stratus clouds. In order to obtain insights about the physical properties of the air masses, the pressure, potential temperature, specific humidity and RH are tracked along the trajectories.

## 3    Spatial distribution of LLC

For the selected stratus and stratus-free nights at Savè supersite, maps of the cloud cover of low clouds at 06:00 UTC at 54 stations across SWA are shown in Fig. 3. Although these reports do not discriminate between stratus and other low cloud types, it is most likely that the reported coverage corresponds to stratus clouds. The information on the low level cloud cover is retrieved from the Karlsruhe African Surface Station Database (KASS-D), which contains long-term, in-situ observations for the whole African continent from various sources. Due to long-standing collaborations with African national weather services and African researchers, KASS-D contains many observations not available in standard, GTS-fed station databases and it is especially valuable source of the precipitation observations (Fink et al., 2017; Vogel et al., 2018). We use these eye observations of cloudiness from trained observers in order to investigate whether the 20 stratus and 6 stratus-free nights at Savè reflect the same cloud structure across the larger regional scale. For both stratus and stratus-free 06:00 UTC observations, Fig. 3 shows lower mean low level cloud cover along the coastal stations as well as farther north of approximately 9°N, while an increased cloud cover is present inland of the coastal strip. Similar distribution of stratiform cloud cover was reported by Schrage and Fink (2012), who analyzed much longer period between May and October 2006. We note that in the central part of the DACCIWA region (Ghana, Togo and Benin) stratus-free nights are characterized with on average lower cloud cover than for stratus nights. For stratus-free nights, the stations in Nigeria and Ivory Coast show on average coverage of more than five octas, which is similar as in stratus nights.

The impact of clouds is also evident in the net longwave radiation measurements near the surface shown for three supersites in Fig. 4. Kalthoff et al. (2018) have found that net longwave radiation of $-10$ W m$^{-2}$ can be considered as a proxy for LLC presence. We note that for stratus nights the median net longwave radiation starts to increase after 23:50 UTC from $-25$ W m$^{-2}$ to $-7$ W m$^{-2}$, which can be related to the occurrence of stratus fractus preceding stratus (Adler et al., 2019). After 02:50 UTC, on most stratus nights LLC are present until sunrise at 06:00 UTC. On the other hand, for stratus-free nights at Savè the net longwave radiation is lower with the median value around $-25$ W m$^{-2}$. The net longwave radiation measurements in Kumasi and Ile-Ife show that stratus nights defined for Savè were also covered with LLC, while stratus-free nights are characterized with lower cloud cover at Kumasi and Ile-Ife as well. Considering that nights with LLC at Savè are characterized with clouds at other locations in the DACCIWA region and the same is observed for stratus-free nights, this suggests that observations of conditions and pocesses leading to the formation of LLC, or the lack of it, at Savè can possible be extended to a larger area.



## 4  Comparison of ABL conditions for stratus and stratus-free nights

### 4.1  ABL conditions at Savè supersite

We investigate the difference in ABL dynamic and thermodynamic conditions during stratus and stratus-free nights by analyzing the composites of the wind speed and potential temperature shown in Fig. 5. At this point the results for Savè presented below are meant to illustrate general similarities and differences in the ABL conditions between nights with and without stratus and not to represent the whole DACCIWA region. Due to the fact that we analyze a small number of stratus-free nights (compared to the number of stratus nights) we present median values, since we notice that one extreme case can considerably alter the mean. Before 18:00 UTC the wind field during stratus nights is characterized by a median wind speed of 4 m s$^{-1}$. For stratus-free nights, starting after 15:00 UTC the wind speed increases from 4 m s$^{-1}$ to a maximum of 7 m s$^{-1}$ in the lowest 750 m a.g.l., indicating an earlier arrival of the maritime inflow. Around 20:00 UTC the onset of a NLLJ is observed for stratus and stratus-free nights with the median wind speed maximum of 8 m s$^{-1}$ at the level of 200 m a.g.l. The height of the jet maximum in stratus nights is increasing with height after 00:00 UTC, which is due to the cloud presence (Adler et al., 2019; Babić et al., 2019), since the static stability below and in the LLC decreases and NLLJ axis shifts upwards towards the top of the layer with higher stability. The jet height for stratus-free nights is approximately constant during the night and is closer to the ground, which is consistent with numerical simulations by Schuster et al. (2013). Additionally, in the second part of the night for stratus-free nights the NLLJ speed can increase further, resulting in stronger wind shear compared to stratus nights (not shown).

With respect to the potential temperature, we note that stratus nights are on average 1 K warmer in the late afternoon and temperature decrease of approximately 4 K over the period of 5 h (18:00–23:00 UTC) is observed in the lowest 700 m. After 00:00 UTC, the potential temperature does not change much mostly due to the presence of the clouds (cf. Babić et al., 2019). In contrast, in stratus-free nights the temperature starts to decrease after 16:00 UTC and, therefore, we observe a cooling by about 2 K in the period between 18:00 and 23:00 UTC, and this cooling is confined to a shallower layer (up to 400 m a.g.l.) than for stratus nights. In the first part of the night, the vertical potential temperature gradient increases until approximately 20:00 UTC in stratus and 22:00 UTC in stratus-free nights, with stronger static stability near the ground between 21:00 and 00:00 UTC for stratus-free nights (not shown). After 00:00 UTC the stability is slightly reduced due to mechanical mixing induced by the increasing wind shear below the jet axis and, additionally, cloud coverage in stratus nights.

Figure 6 shows that during the afternoon the near-surface wind speed and turbulent kinetic energy (TKE) are higher during stratus-free nights than in stratus nights, while after sunset both have low values (Fig. 6a, d). Lothon et al. (2008) found that NLLJ can induce dynamical turbulence down to the surface and the impact of the NLLJ onset is visible in the near-surface measurements (Fig. 6a, d). While UHF profiler measurements indicate concurrent onset of NLLJ (Fig. 5), the near-surface wind speed and TKE increase 1 h later in stratus-free nights compared to stratus nights. This is most likely related to the stronger static stability close to the surface in stratus-free nights between 18:00 and 22:00 UTC (not shown). The NLLJ causes an increase in turbulent mixing close to the surface, which for stratus nights on average does not change much during the whole night, mostly due to the presence of LLC (Fig. 6d). Differences in the near-surface temperature between stratus and stratus-





free nights are only a few degrees in the late afternoon (Fig. 6b), with on average higher temperatures for stratus-free nights, however, the large range of variability is also present. The temperature decreases during the whole night for stratus-free nights, whereas the impact of LLC is reflected in nearly constant near-surface temperature after 00:00 UTC for stratus nights. Due to the radiative cooling, which leads to the build-up of stable stratification close to the surface in stratus-free nights, the decrease

of wind and turbulence is observed after 01:00 UTC (Fig. 6a, d). The distinct difference between stratus and stratus-free nights is evident in specific humidity, which is on average around 1 g kg$^{-1}$ lower for stratus-free nights (Fig. 6c).

## 4.2 ABL conditions in the DACCIWA region

In order to analyze the ABL conditions in the DACCIWA region, we compare vertical profiles of wind speed, potential temperature and specific humidity from radiosondes measured at 18:00 and 00:00 UTC at Savè, at three coastal stations (Abidjan,

Accra and Cotonou) and at Lamto and Parakou during stratus and stratus-free nights. At Savè unfortunately just one profile for stratus-free nights is available, corresponding to one stratus-free IOP, while the stratus-free profiles in Lamto and Parakou reflect conditions during the vortex period in mid-July. The number of available radiosonde profiles included in the analysis at each station is given in Tab. 1.

Figure 7 shows wind speed profiles at 18:00 and 00:00 UTC. For the investigated period, the mean wind conditions at the

coastal stations are similar for stratus and stratus-free nights, with similar variability, suggesting quite stationary monsoon flow. Meridional and zonal wind components indicate slightly stronger southerly flow for stratus nights and only slightly stronger westerly flow on clear nights (not shown). The ABL wind profiles for stratus and stratus-free nights at Savè supersite at 18:00 UTC are characterized with a mean wind of 4 m s$^{-1}$ and a large standard deviation, which is a result of a combination of days with weak wind and days with strong monsoon flow (Adler et al., 2019; Dione et al., 2018). The large wind speed standard

deviations are observed at two other inland stations Lamto and Parakou, with stronger winds at the northernmost station Parakou. Later at 00:00 UTC, a NLLJ wind profile is observed at three inland stations during both stratus and stratus-free nights. The magnitude of the mean wind at Lamto and Savè corresponds to the wind speed observed at the coast. Farther north at Parkou the average wind speed has even larger values than at coastal stations, however, a substantial variability is observed as well. At Savè supersite for this particular stratus-free night wind speed has a maximum below 250 m a.g.l. corresponding to

the top of the shallow inversion layer (not shown). The NLLJ in this particular case arrived after 00:00 UTC, as shown in Sect. 5, and the same was observed at Parakou.

The potential temperature profiles at 18:00 and 00:00 UTC indicate similar values for stratus and stratus-free nights at three coastal stations and in Lamto (not shown). More pronounced differences in potential temperature are observed at Savè and Parakou at 18:00 UTC, with an average temperature below 750 m a.g.l. for stratus nights roughly 1 K higher than for the

stratus-free nights. At 00:00 UTC, the stratus and stratus-free profiles are quite similar, with differences in the lowest 500 m a.g.l. on the order of 0.5 K, which is within the measurement accuracy of the sensor. Between 18:00 and 00:00 UTC, on average only a slight decrease in potential temperature occurred at the coast, indicating the presence of the same maritime air mass (not shown). For the inland stations Savè and Parakou, the temperature for stratus nights decreased on average by 3 K below 500





m a.g.l., while in stratus-free nights temperature decreased on average by 1.5 K in the same layer, leading to a much weaker cooling rate during these 6 h.

The largest differences between stratus and stratus-free nights are observed in specific humidity not just at Savè, but also at all other stations (Fig. 8). That is, for stratus-free nights specific humidity is on average 1 g kg$^{-1}$ lower at six radiosonde stations, while for individual cases differences at Savè go up to 4 g kg$^{-1}$ in the late afternoon hours. This implies that the background level of specific humidity potentially has an important role in LLC formation. Possible reasons for the reduced specific humidity are discussed in Sect. 7.

## 5   Comparison of atmospheric conditions and quantification of RH tendency for two contrasting cases

In order to asses the impact of the observed differences in ABL conditions on processes relevant for LLC formation, we investigate in detail two IOP cases: IOP 7 (4–5 July), which is a rather typical stratus night case (Adler et al., 2019), and IOP 10 (13–14 July), which represents a stratus-free night. For the investigation of atmospheric conditions and processes during these IOPs, we use normal and frequent radiosondes, which were released with a temporal resolution of 1.5 h starting at 17:00 UTC.

In stratus case the period between 18:00 and 00:00 UTC is characterized by a quite strong southwesterly monsoon flow and an early onset of NLLJ, already around 18:00 UTC (Fig. 9a). Prior to LLC formation at 00:00 UTC, the potential temperature indicates a strong temperature decrease and specific humidity values are between 16 and 19 g kg$^{-1}$ in the lowest 1000 m a.g.l. (Fig. 9c). On the other hand, the same period during stratus-free case has much lower RH (Fig. 9d), especially in the layer between 100 and 700 m a.g.l., with westerly and northwesterly wind which is weaker than the average for status-free nights (Fig. 9b). The specific humidity reaches values between 14 and 16 g kg$^{-1}$ in the lowest 1000 m a.g.l., i.e. it is 2–3 g kg$^{-1}$ lower than in stratus night. Around 23:00 UTC, we notice a wind direction change to southerly, which is accompanied by stronger decrease in temperature over a 600 m deep layer, together with an increase in specific humidity of about 2 g kg$^{-1}$ up to 17 g kg$^{-1}$ and substantial increase in RH during the second part of the night, however, the saturation is not reached (Fig. 9b, d). In the stratus case, LLC form around 00:00 UTC, with CBH at 250–300 m a.g.l. and RH profiles indicate on average a 250–300 m deep cloud layer. The conditions observed afterwards are very similar to conditions observed on another typical IOP (IOP 8, Babić et al., 2019).

Figure 10 shows the contributions from observed temperature and specific humidity changes to the RH tendency for these two case studies (the formula is given in Babić et al., 2019). These contributions are calculated from the radiosonde measurements at 17:00, 23:00 and 05:00 UTC. Figure 10a shows the period between 17:00 and 23:00 UTC and evidently for a stratus case a strong increase in RH, with a total change of around 25 % below 500 m a.g.l., is observed mostly due to cooling, while the contribution from moistening is only about 5 %. This is in correspondence with the results for 11 IOPs in Adler et al. (2019). At the same time, RH change during stratus-free night is much weaker, with both terms contributing equally. In the second part of the stratus night, RH tendency is close to zero, mostly due to opposite contributions from cooling and drying during this period (Fig. 10b). For the stratus-free night, an increase in RH after 00:00 UTC is observed mostly due to the cooling, while the specific





humidity contribution is negligible below 500 m a.g.l. The stronger cooling in the second part of the clear night is most likely related to the wind direction change to southerly and the onset of the NLLJ after 23:00 UTC. However, the maximum increase of RH below 500 m a.g.l. is about 15 %. Previous studies have shown that the cooling during the period of southwesterly NLLJ onset is mostly caused by the horizontal advection of cold maritime air mass (Adler et al., 2019; Babić et al., 2019).

Therefore, using the approach described in Adler et al. (2019) we estimate the contribution from horizontal cold-air advection to the observed cooling. For the stratus case, the horizontal advection is estimated using radiosonde measurements from 17:00 and 23:00 UTC, while for stratus-free case it is estimated from 23:00 and 05:00 UTC measurements (since the onset of NLLJ in the stratus-free case is observed after 23:00 UTC) and the results are shown in Fig. 10c. We note that for the stratus case the mean cooling rate in the layer 200–400 m a.g.l., i.e. the level of NLLJ axis, is on the order of $-0.4$ K h$^{-1}$. One the other hand,

a weak cooling in the second part of stratus-free night is a result of a smaller contribution from the cold-air advection, which has the maximum mean value of $-0.2$ K h$^{-1}$. The relation of the NLLJ onset time to the monsoon strength and its implications on the formation of LLC are discussed in Sec. 7.

## 6 The influence of large-scale conditions on the formation of stratus clouds

Using ERA5 reanalysis, we now compare large-scale conditions during stratus and stratus-free nights focusing on the con-

ditions in the boundary layer. Our aim here is to investigate whether the observations during the DACCIWA campaign are reproduced by reanalysis and, consequently, to examine the representability of these conditions for the whole SWA. The investigation of large-scale conditions should also elucidate where the air masses, which cause drier conditions during stratus-free nights, come from.

Figure 11 shows geopotential height, wind, temperature and specific humidity differences between stratus and stratus-free

nights at 18:00 UTC at 950 hPa isobaric level, which is approximately in the middle of the ABL and also corresponds to the mean stratus layer (e.g., Kalthoff et al., 2018; Dione et al., 2018). Northwest of the DACCIWA region, in the region that corresponds to the Saharan Heat Low (SHL), lower geopotential heights of the 950 hPa isobaric level are evident for stratus nights, indicating slightly stronger SHL than in stratus-free nights. The overall lower pressure in stratus-free nights over the whole area of investigation is mostly due to the low pressure during the vortex period. Also, overall lower pressure is more

often observed in the pre-monsoon period (Schrage et al., 2007; Knippertz et al., 2017).

With respect to the wind speed, we find a quite good agreement between the observations and ERA5 data, which show slightly stronger wind at inland stations during stratus nights, while differences in Abidjan and Cotonou are very small in both observations and reanalysis (Fig. 11b). The large differences in wind speed between stratus and stratus-free nights measured for Accra are not obvious in ERA5 data. A possible reason for this are the two missing radiosonde measurements in June. We

note that during stratus nights a slight reduction in wind speed exists along the coast and in the Gulf of Guinea. Differences in the wind direction suggest that this could be related to slightly stronger westerly flow component on clear nights, which also indicate slightly stronger southerly wind component over the ocean in stratus nights. A band of higher wind speeds around 6 and 7 °N is also visible, probably related to the stronger maritime inflow front winds during stratus nights. The differences in





the wind direction over the land, which do not show any preferred direction, are most likely the result of mesoscale convective systems in the area.

In the reanalysis data, stratus nights in central parts of the DACCIWA region are on average colder (up to 1 K) than stratus-free nights (Fig. 11c). This is contrary to observations at Savè (Figs. 5a, b) and for Parakou (not shown), which are, according to

measurements, on average 1 K warmer during stratus nights. At other stations radiosonde measurements indicate differences in temperature on the order of 0.5 K (not shown) and the same differences, but of the opposite sign, are found in ERA5. However, these differences are within the range of the instrument measurement accuracy. In the whole SWA region stratus-free nights are drier compared to stratus nights (Fig. 11d), with a difference on the order of 1 g kg$^{-1}$, which is consistent with the observations (Fig. 8), and the strongest signal is seen in the central part of the DACCIWA region.

Figure 12a, b show each of the trajectories for stratus and stratus-free nights started at 03:00 UTC, since at this time stratus clouds form for the most stratus nights, integrated backward for 48 h and averaged between 30 and 50-hPa above the ground, i. e. the layer corresponding to the mean cloud layer. The starting time for each trajectory corresponds to 0 h. We note that trajectories in stratus nights on average stretch farther above the Atlantic ocean and their travel time over the land is shorter (14.7 h on average) than for stratus-free nights (17.2 h on average). For four out of six analyzed stratus-free trajectories, the

origin is around 4°W, while the two slowest trajectories (closer to the coast) start around 0°E. Generally, there are no substantial differences in the origin and path of the trajectories for stratus and stratus-free nights and the air parcels travel with the dominant southwesterly monsoon flow.

The thermodynamic history of the air parcels which represent the near-surface air mass during stratus and stratus-free nights is shown in Fig. 13. In addition to median values of pressure, RH, specific humidity and potential temperature along the

trajectories, the individual backward trajectories for each stratus and stratus-free night are shown, since a substantial variability for each of the parameters is present in both cases, which are more pronounced for stratus-free nights. The temporal evolution of the pressure along the trajectories indicates that air parcels in stratus and stratus-free nights are on average found within the marine boundary layer before reaching the land. Analysis of individual trajectories indicates that the two trajectories (during two consecutive nights in June, Fig. 12b) originated from levels above 900 hPa, that is, well above the marine boundary layer

with potential temperature above 300 K and specific humidity below 11 g kg$^{-1}$ at −48 h (Figs. 13a, c, d). These two June cases are rather different than the other four stratus-free nights, therefore, affecting the average statistics, as clearly seen in the RH, specific humidity and potential temperature medians. Unaffected by these two extreme cases is the temporal evolution of specific humidity along the trajectories, which indicates its increase and moistening of air parcels as they approach Savè, and the same is found for stratus nights (Fig. 13c).

In addition to the statistical overview, temporal evolution of pressure, relative and specific humidity, and potential temperature for trajectories corresponding to stratus and stratus-free cases analyzed in Sect. 5 illustrates that air parcels in the stratus-free case can originate and travel within the marine boundary layer with similar specific humidity and temperature as in the stratus case (Fig. 13). However, during the stratus-free case air parcels reach the coastline already during the previous night (Fig. 12b) and, therefore, exhibit stronger warming in the convective boundary layer over land between 09:00 and 15:00 UTC

(Fig. 13d). This results in a strong decrease of air parcel's RH with a minimum around 15:00 UTC, which afterwards increases





linearly up to 94 % until 03:00 UTC (Fig. 13b). On the other hand, for the stratus case the RH is nearly 100 % at 03:00 UTC, which is consistent with observations, while the specific humidity is only 0.5 g kg$^{-1}$ higher and the potential temperature is 0.5 K lower than in the stratus-free case. This illustrates the complexity of atmospheric conditions and involved processes over SWA. Namely, stratus-free nights occur in situations when air parcels originate above the marine boundary layer and, thus,

are drier and warmer, or in cases when they experience a strong warming in the convective boundary layer due to the longer presence over the land, while at the same time being slightly drier.

## 7 Discussion

In this section we compare our results with previous studies on this topic, and discuss possible pathways for the reduced moisture during stratus-free nights. For instance, Schrage et al. (2007) also found drier conditions for clear nights at all levels

farther north of Savè at Parakou. They state that the differences in moisture profiles in the middle and higher troposphere reflect differences in large-scale monsoon conditions. Additionally, they found a presence of dry layers for clear nights corresponding to layers of northerly flow and high dew point depression. In our case, the observed differences in specific humidity are limited to the lowest 2 km a.g.l. for Savè data and without dry layers present (not shown), while pronounced differences at coastal stations are up to 3.5 km with individual dry layers between 1.5 and 3 km (not shown). A question which arises is where

does this dry air come from? We note that 50 % of the stratus-free nights were observed during mid-July when a propagating cyclonic–anticyclonic vortex couplet occurred (Knippertz et al., 2017). A southern anticyclonic system slowly propagated from Gabon across the tropical eastern Atlantic, bringing dry air from the area of subsidence in the equatorial zone or even southern hemisphere (Knippertz et al., 2017). The radiosondes from Abidjan show a sudden intrusion of very dry air into the 850–700 hPa layer on 11 July causing a drop of RH from 85 to under 20 % and persisting until the morning of 14 July (Knippertz

et al., 2017). In the central part of this episode, which occurred between 22:00 UTC on 11 July and 22:00 UTC on 13 July, a drop of specific humidity to only 3 g kg$^{-1}$ was recorded. The same dry layers between 1.5 and 3 km a.g.l. are observed for the other two coastal stations, with slightly higher values indicating that the western parts of the Guinean coast were more strongly affected (not shown). These dry layers most likely originated from a central and southern African dry air mass, which is typically filled with biomass burning aerosol, and is transported to the west over the eastern tropical Atlantic in the layer

between 2–4 km a.g.l. (Haslett et al., 2019). Haslett et al. (2019) have found that a considerable proportion of the biomass burning aerosol (smoke) is entrained into the marine boundary layer and advected northwards with the southerly trade winds towards the Coast of Guinea. This conclusion is supported by the evidence of aged aerosol particles which were measured by a research aircraft (Flamant et al., 2018; Haslett et al., 2019). These studies suggest that reduced moisture over West Africa is a result of intrusions of dry air originating either from the Southern Hemisphere (with the biomass burning aerosol as a tracer)

in its southern parts (Haslett et al., 2019) or from dry Saharan air layer in its northern parts (Schrage et al., 2007).

Our results are contrary to Schrage et al. (2007) and Schuster et al. (2013), who found stronger NLLJ for cloudy nights as a consequence of a stronger monsoon circulation. Possible reasons why we do not observe stronger NLLJ and stronger monsoon for stratus nights at Savè supersite could be due to the investigated period between mid-June and end of July, which





mostly corresponds to post-onset phase of the monsoon (Knippertz et al., 2017). Namely, the onset of West African monsoon flow is related to the surface pressure contrast between the SHL, which develops due to the intense surface heating during boreal summer (Lavaysse et al., 2009), and relatively cool waters of the eastern Tropical Atlantic. Typically, the monsoon occurs around the end of June (Janicot et al., 2008; Sultan and Janicot, 2003), while during 2016 its onset occurred on 22 June

(Knippertz et al., 2017). Therefore, the differences in the NLLJ for cloudy and clear nights found in previous work could be related to the analysis of larger number of nights from the pre-monsoon or late monsoon period compared to our analysis. Another possible reason is perhaps due to the different geographic location analyzed, since the characteristics of NLLJ can differ across the region (Schuster et al., 2013). On the other hand, Schrage et al. (2007) have concluded that the differences between cloudy and clear nights are most likely related to day-to-day synoptic changes in the monsoon structure and the similar

conclusion seems to apply to our analysis.

   Furthermore, the analysis of the two cases studies suggests that the magnitude of the cooling related to the cold-air advection with the maritime inflow depends on the timing of the NLLJ onset and its speed. Dione et al. (2018) found the timing of the maritime inflow to be correlated with the strength of the monsoon flow and, therefore, late arrival of the NLLJ during stratus-free case study suggests weaker monsoon flow. However, this one case does not necessarily reflect the general conditions for

all clear nights. Schuster et al. (2013) also found that stronger cold-air advection up to about 1000 m supports cooling during cloudy nights. Hence, it seems that in our case the combination of weak cooling due to the smaller contribution from horizontal cold-air advection and low background values of moisture resulted in a stratus-free night.

## 8   Summary and conclusions

This study investigates differences in the boundary layer conditions between stratus and stratus-free nights over SWA during

the summer monsoon season. We used observational data collected during the DACCIWA campaign as well as the newest ERA5 data set. Conditions during 6 stratus-free nights are compared with 20 nights with LLC using in-situ and remote sensing measurements at Savè supersite and radiosonde measurements from coastal stations (Abidjan, Accra and Cotonou) and inland stations in Lamto and Parakou. The analysis is complemented with investigation of the large-scale conditions during the selected nights, for which we use ERA5 reanalysis, as well as with backward trajectories calculation using LAGRANTO.

We find that LLC typically form during undisturbed monsoon flow, while stratus-free nights occured in the pre-monsoon period as well as after the monsoon onset. This implies that the differences between stratus and stratus-free nights are most likely a result of day-to-day synoptic changes within the monsoon flow. For example, the three consecutive stratus-free nights were observed in mid-July during the vortex period, when a cyclonic-anticyclonic vortex couplet passed across the area of investigation (Knippertz et al., 2017). Considering the small number of stratus-free nights observed, we note that the conclusions

on general conditions and similarities or differences between stratus and stratus-free nights have to be drawn carefully. Hence, we pay a special attention to possible extreme cases which might influence the general conclusion.

   The onset of NLLJ at Savè supersite characterizes both stratus and stratus-free nights, with similar onset time and strength. With the onset of NLLJ a shear-induced turbulent mixing is observed in both cloudy and clear nights, which is further main-





tained after the LLC form in stratus nights, since the LLC presence weakens the surface cooling. For stratus-free nights, the near-surface layer continues to cool and, simultaneously, the wind speed and turbulent mixing start to decrease. Around the sunset at 18:00 UTC stratus nights are on average 1 K warmer than stratus-free nights with similar temperatures around 00:00 UTC, which corresponds to slightly stronger cooling in the layer between 100 and 700 m a.g.l. during stratus nights.

5 Furthermore, the near-surface measurements of specific humidity at Savè and from radiosondes at additional five locations in the DACCIWA region suggest that stratus-free nights are on average 1 g kg$^{-1}$ drier than stratus nights. These decreased values of humidity are observed during the vortex period in mid-July, when a southerly anticyclonic vortex, originating from Gabon in Central Africa, transported a dry air mass into the DACCIWA region (Knippertz et al., 2017; Haslett et al., 2019). Considering that stratus did not form during three consecutive nights in the vortex period, this implies the potentially important
10 role of the background level of specific humidity on the LLC formation. In addition to this, the analysis of two contrasting case studies, presented in Sect. 5, shows that the timing and onset of NLLJ, which is related to the arrival of maritime inflow, may affect the horizontal cold-air advection. Namely, for the clear night case with a late NLLJ onset (after 23:00 UTC) the weak magnitude of cooling is mostly due to the small contribution by horizontal cold-air advection, which equals to half of the typically observed cooling rate (cf. Adler et al., 2019). Finally, the backward trajectories analysis suggests that stratus-free
15 nights can happen in cases when air parcels reaching Savè supersite are either originating from drier levels above the marine boundary layer or when they spend longer time over the land, experiencing stronger warming in the convective boundary layer, which leads to stronger decrease of RH than for stratus nights.

 Similarly as in the modeling study of Schuster et al. (2013), our results show that subtle differences in the strength and onset time of the maritime inflow and NLLJ, in cooling rates mostly due to the horizontal cold-air advection and, additionally, in
20 background specific humidity discriminate whether LLC will form or not. Please note that not necessarily all of these processes are simultaneously occurring for each of the clear nights.

 These results highlight that in order to adequately simulate the LLC over SWA it is crucial to reproduce and capture the main features, i.e. the onset time and strength of the maritime inflow and NLLJ, turbulent mixing, cold-air advection and the background moisture level. Considering that the involved processes range from synoptic and mesoscale to the boundary layer
25 microscale, it is necessary to apply numerical models with horizontal grid spacing able to resolve the involved scales of motions and to use appropriate parameterization schemes, respectively. Finally, the comprehensive, high-quality DACCIWA data set can serve as an observational benchmark for investigating which of the relevant processes are realistically reproduced in the numerical models as well as to contribute to their improvement.

*Data availability.* The DACCIWA data from Savè supersite and coastal radiosonde stations are available on the SEDOO database (http:
30 //baobab.sedoo.fr/DACCIWA/; Derrien et al., 2016; Handwerker et al., 2016; Kohler et al., 2016; Maranan and Fink, 2016; Wieser et al., 2016) for scientists interested in boundary-layer studies in southern West Africa.



*Author contributions.* KB wrote the manuscript with contributions from all co-authors. The data analysis was mostly conducted by KB and to smaller extent by BA and JFQ. BA, NK, CD, FL and ML participated in the DACCIWA field campaign.

*Competing interests.* The authors declare that they have no conflict of interest.

*Acknowledgements.* The DACCIWA project has received funding from the European Union Seventh Framework Programme (FP7/2007-2013) under grant agreement no. 603502. The contribution of JFQ was supported by the Helmholtz Association (grant VH-NG-1243). We thank the staff of KIT (Karlsruhe Institute of Technology) and UPS (University of Toulouse) for helping to install and run the equipment as well as to staff of INRAB in Savè for allowing to use their grounds for the experiment. We thank Andreas Fink for fruitful discussions and Anke Kniffka for providing the KASS-D data.





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





**Table 1.** The number of radiosonde profiles available at 18:00 and 00:00 UTC at coastal stations Abidjan, Accra, Cotonou and stations inland of coastal strip Lamto, Savè and Parakou during stratus and stratus-free nights.

|  | Abidjan | Accra | Cotonou | Lamto | Savè | Parakou |
|---|---|---|---|---|---|---|
| **18:00 UTC** |  |  |  |  |  |  |
| stratus | 19 | 14 | 11 | 7 | 14 | 13 |
| stratus-free | 3 | 4 | 4 | 3 | 1 | 3 |
| **00:00 UTC** |  |  |  |  |  |  |
| stratus | 20 | 13 | 12 | 4 | 10 | 14 |
| stratus-free | 5 | 4 | 5 | 3 | 1 | 3 |





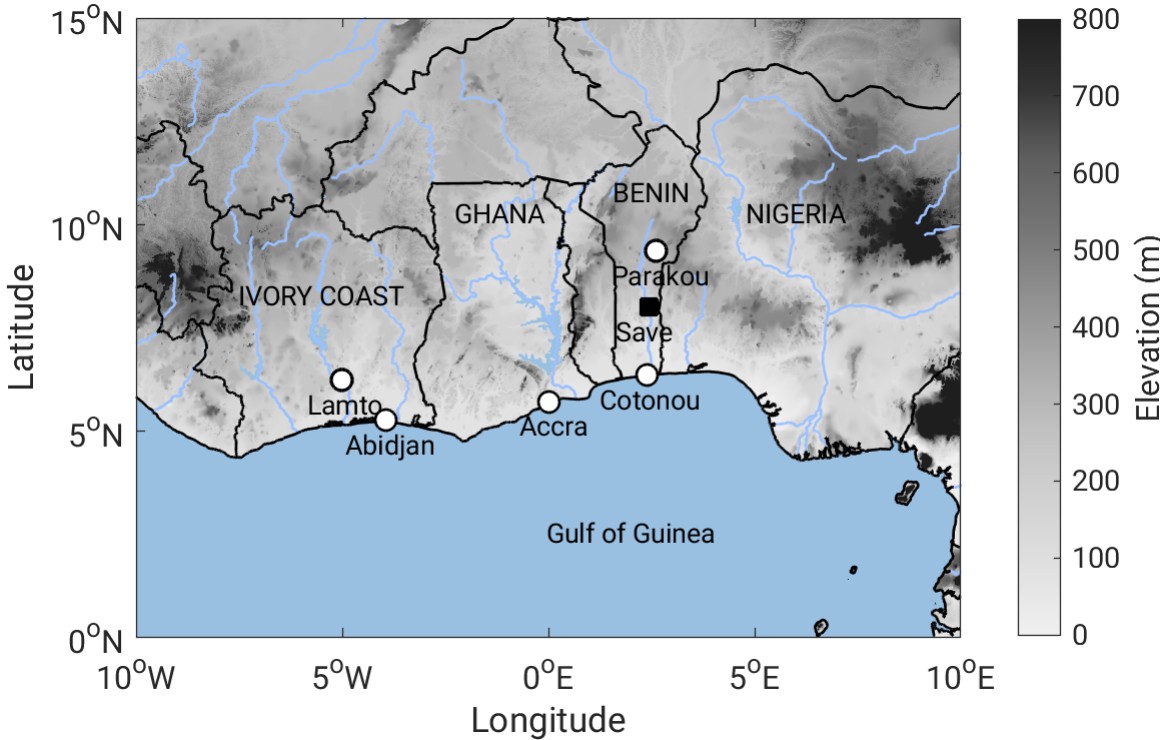

**Figure 1.** Geographical map of the DACCIWA study region. The location of the highly instrumented Savè supersite is indicated with black square. The white dots indicate the location of radiosonde measurements performed as part of the radiosonde campaign (Maranan and Fink, 2016) along the coast in Abidjan (Ivory Coast), Accra (Ghana) and Cotonou (Benin) and inland at Lamto (Ivory Coast) and Parakou (Benin).

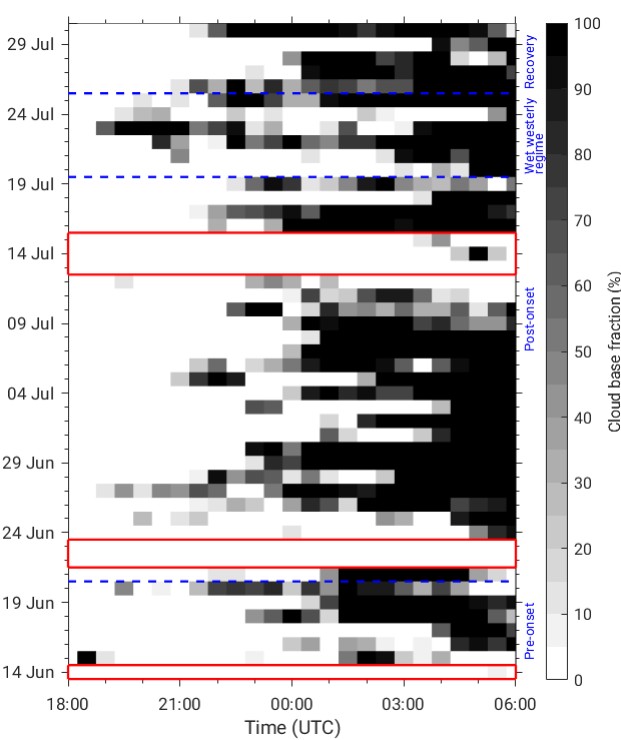

**Figure 2.** The cloud-base fraction for CBH detected between 0 and 500 m a.g.l. measured at Savè in the period between 18:00 and 06:00 UTC during the whole DACCIWA campaign. Red rectangles indicate 6 stratus-free nights. The periods of different synoptic phases (Knippertz et al., 2017) are denoted with blue dashed lines.



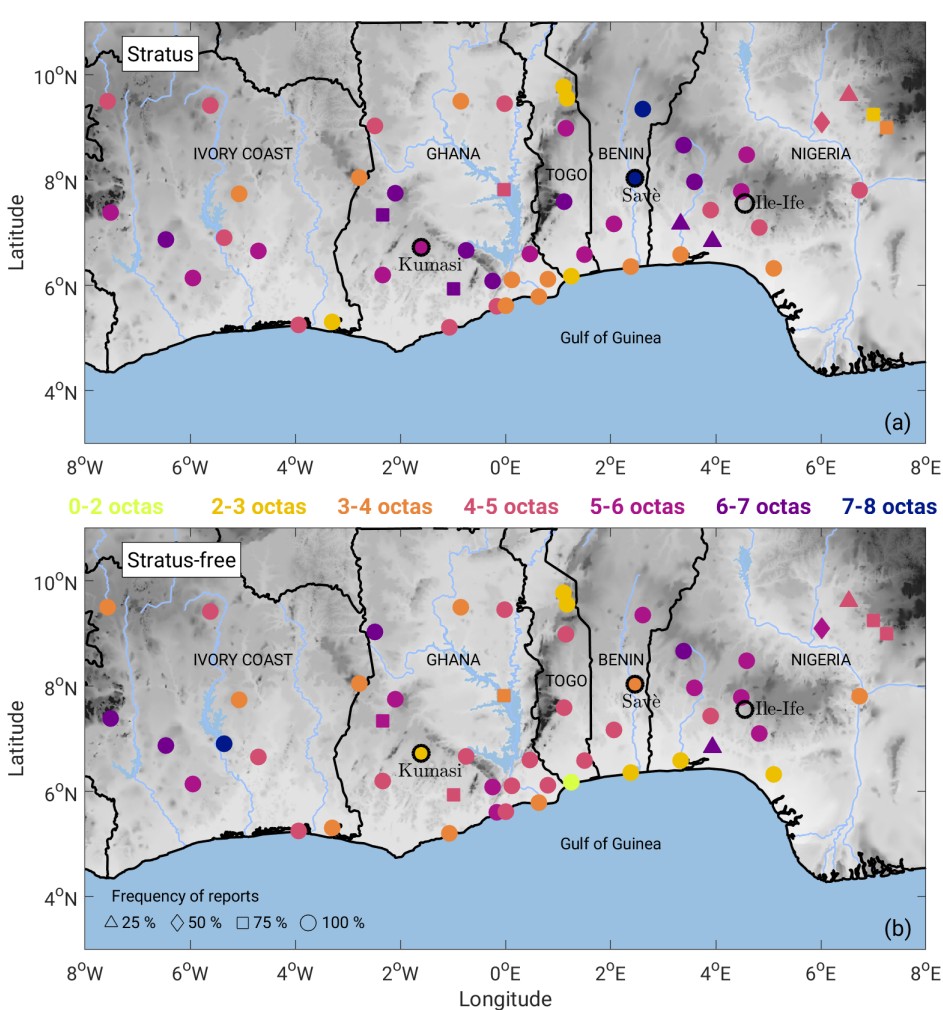

**Figure 3.** The mean low level cloud coverage (color) reported at synoptic weather stations across SWA at 06:00 UTC for stratus **(a)** and stratus-free **(b)** nights. Different marker types indicate how often the reports were available at a particular station.

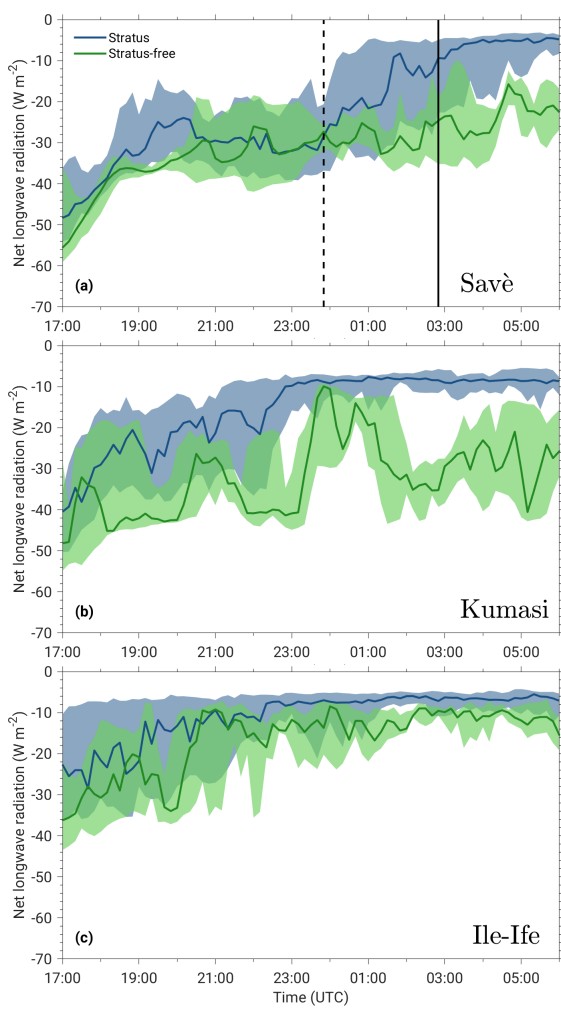

**Figure 4.** The net longwave radiation for stratus (blue) and stratus-free (green) nights measured at three superistes: **(a)** Savè, **(b)** Kumasi and **(c)** Ile-Ife. The solid line shows the median and the shading indicates the range between 25th and 75th percentile.

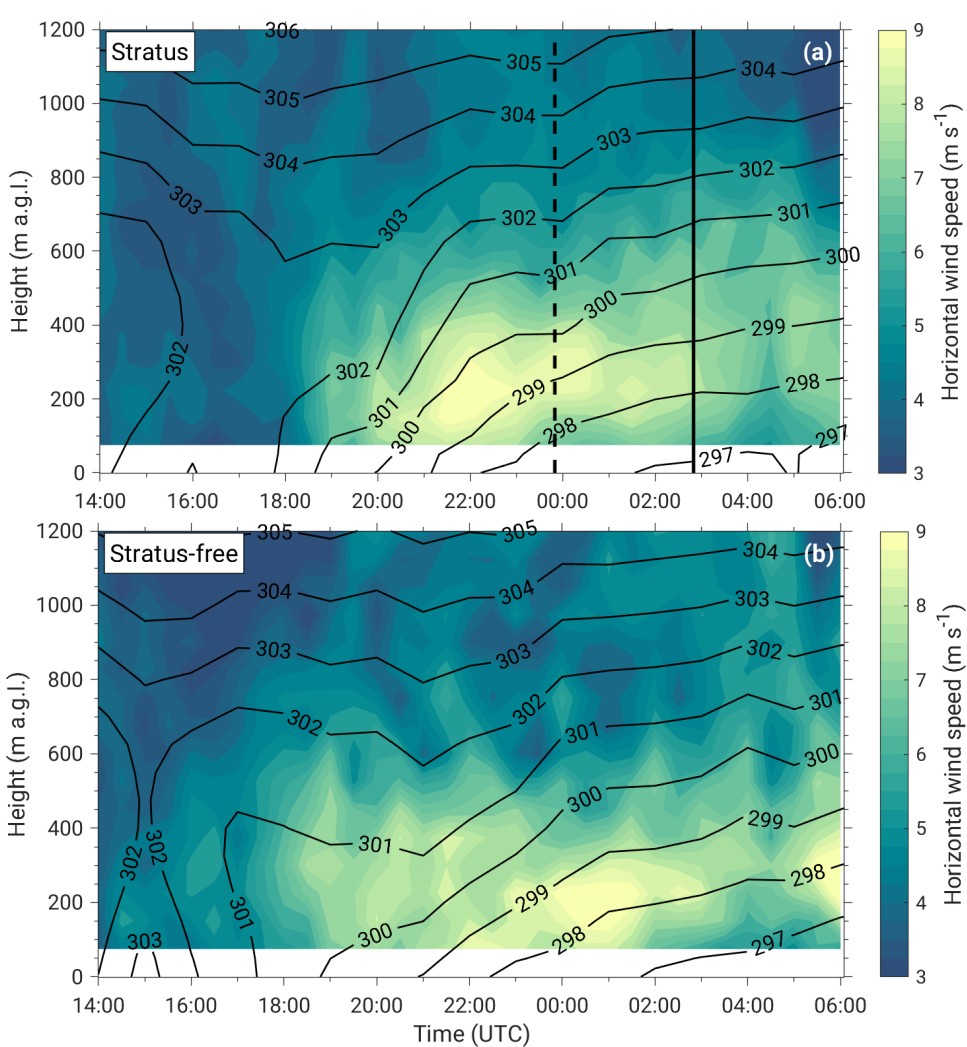

**Figure 5.** Time-height median composite of horizontal wind speed (color) and potential temperature (black isolines) for stratus **(a)** and stratus-free **(b)** nights.



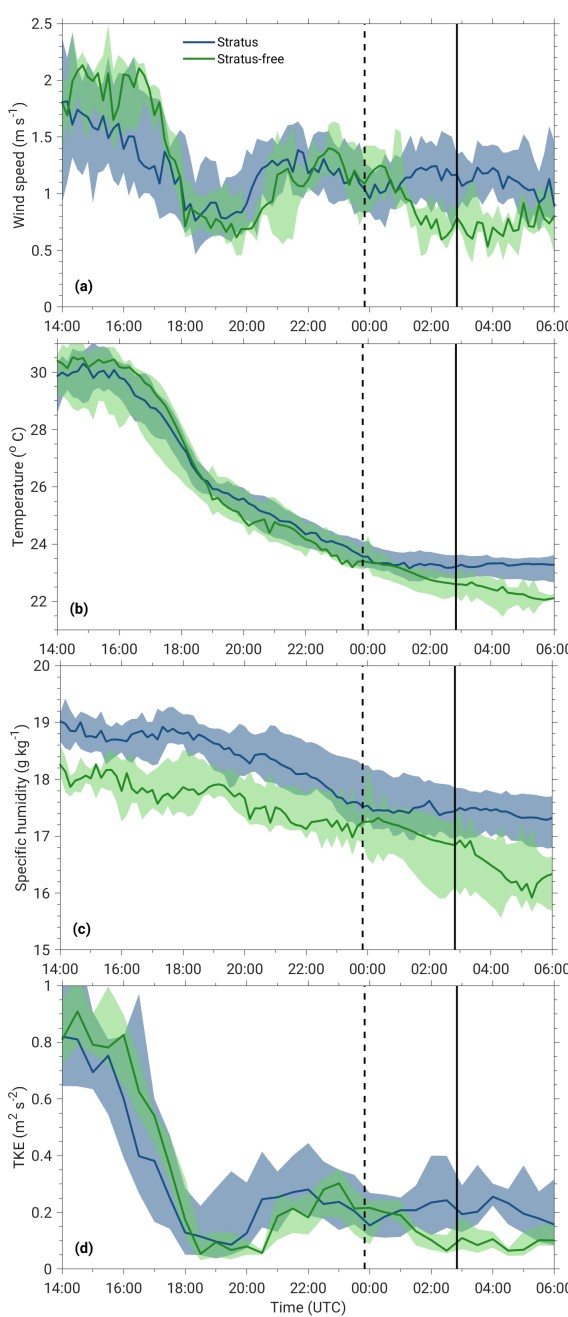

**Figure 6.** Near-surface measurements (4 m a.g.l.) of temperature **(a)**, wind speed **(b)**, specific humidity **(c)** and turbulent kinetic energy (TKE, **d**) at Savè. The solid line indicates the median and shaded area represents the range between 25th and 75th percentile. Stratus and stratus-free nights are shown in blue and green color, respectively.



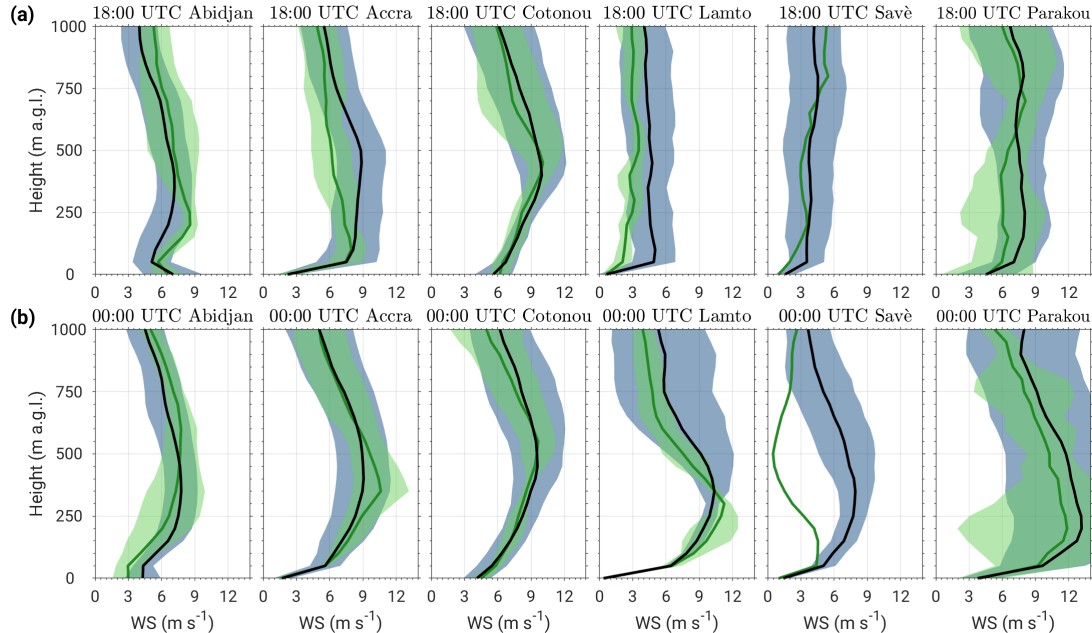

**Figure 7.** Vertical profiles of horizontal wind speed (WS) measured with radiosondes at 18:00 UTC **(a)** and 00:00 UTC **(b)** at inland stations Lamto, Savè and Parakou and coastal stations Abidjan, Accra and Cotonou. The solid line indicates the mean and shading represents the standard deviation. Stratus and stratus-free nights are shown in blue and green color, respectively. The number of analyzed profiles for each station is given in Tab. 1.

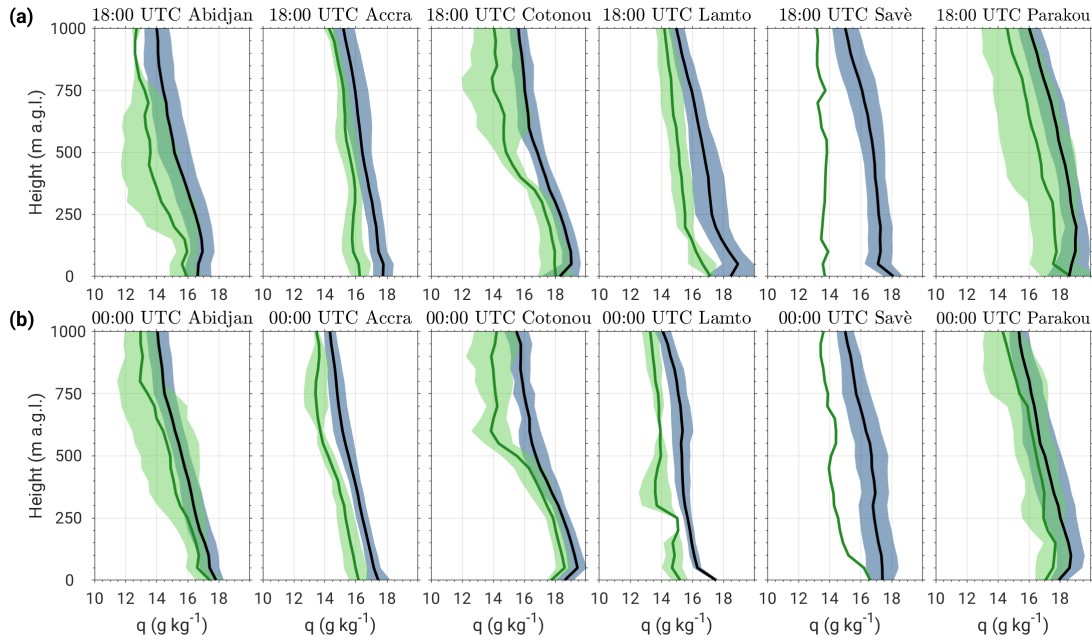

**Figure 8.** The same as Fig. 7, but for specific humidity.

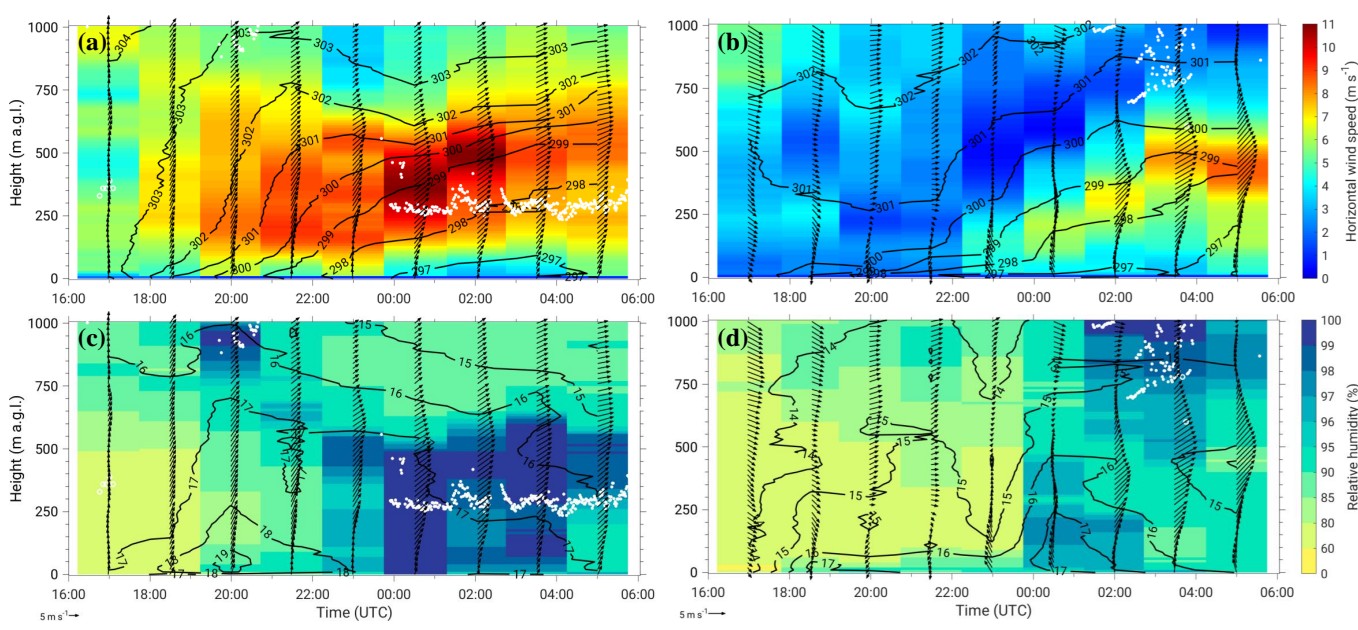

**Figure 9.** Temporal evolution of horizontal wind speed (color), horizontal wind vectors (arrows) and potential temperature (black isolines) measured with normal and frequent radiosondes every 1.5 h during stratus case (IOP 7, **a**) and stratus-free case (IOP 10, **b**). The white dots indicate the CBH. Temporal evolution of RH (color) and specific humidity in g kg$^{-1}$ (black isolines) for IOP 7 (**c**) and 10 (**d**).

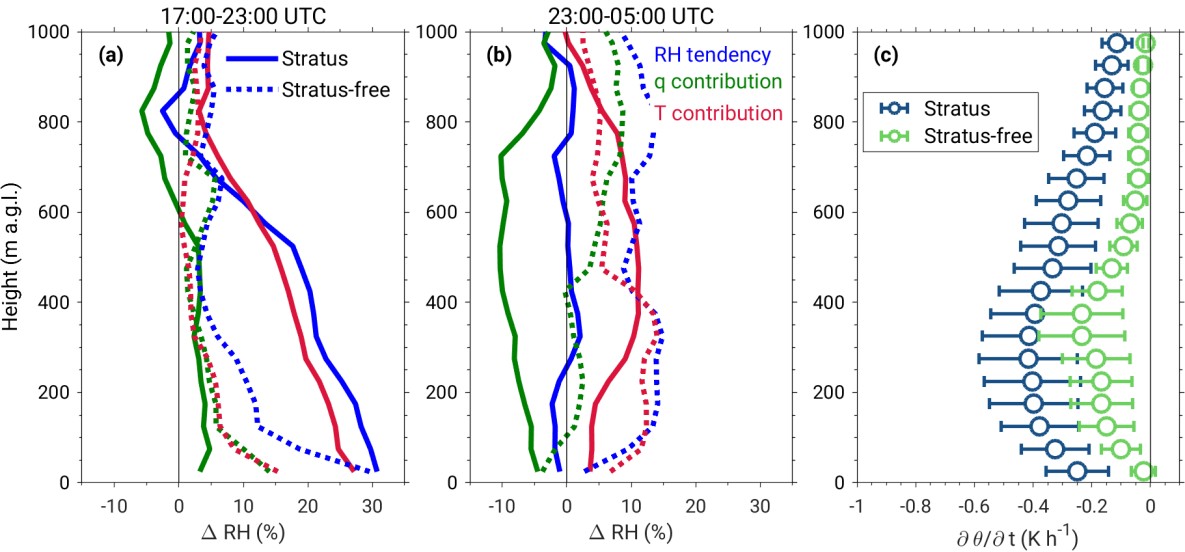

**Figure 10.** The observed RH tendency profiles obtained from radiosonde measurements for the period 17:00–23:00 UTC **(a)** and 23:00–05:00 UTC **(b)** are shown in blue. The contributions from specific humidity changes and temperature changes are shown in green and red, respectively. Solid lines represent the stratus case (IOP 7) and dotted lines show stratus-free case (IOP 10). **(c)** Mean profiles of the contributions by horizontal advection to the observed cooling for the period 17:00–23:00 UTC for stratus case (blue) and for the period 23:00–05:00 UTC in stratus-free case (green). Error bars indicate the standard deviation, which is estimated from the calculations for three coastal stations (Abidjan, Accra and Cotonou) and four different maximum inland propagation distances of the maritime inflow front (50, 75, 100 and 125 km).

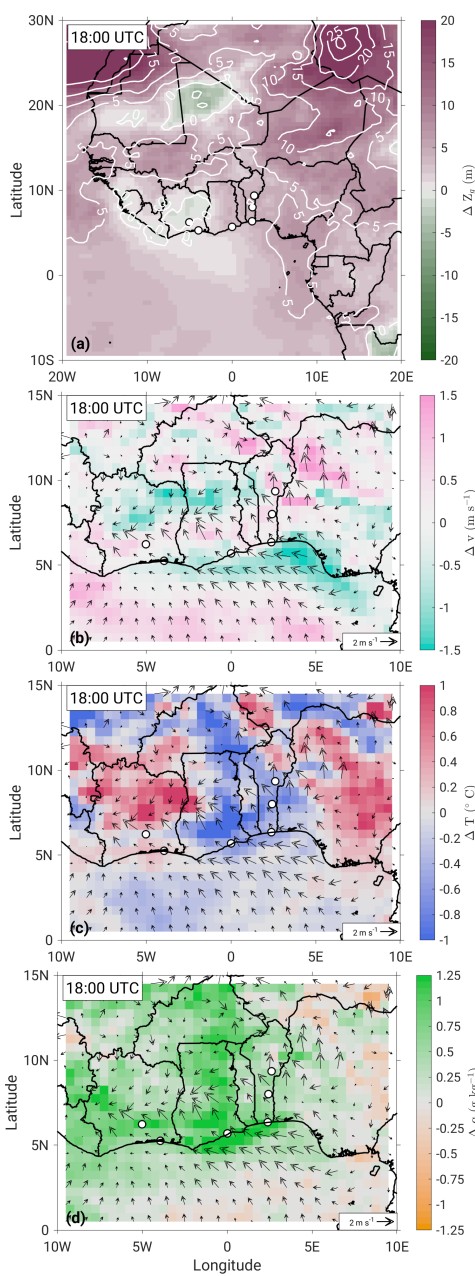

**Figure 11.** The differences in **(a)** geopotential height (color and white isolines), **(b)** horizontal wind speed (color) and wind vectors (arrows), **(c)** temperature and **(d)** specific humidity between stratus and stratus-free nights at 950 hPa isobaric level at 18:00 UTC from ERA5 reanalysis.



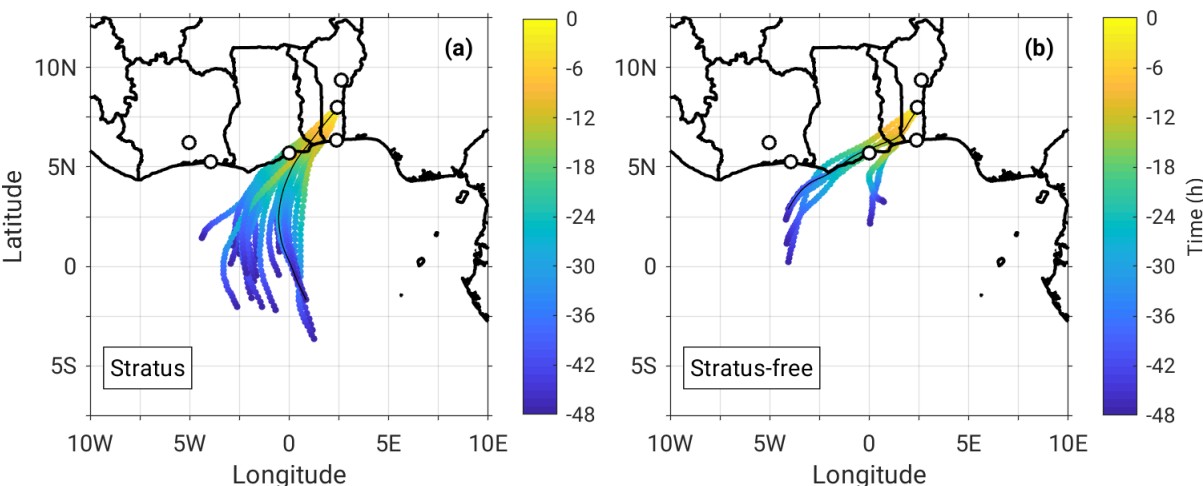

**Figure 12.** Geographical representation of the mean trajectories in the layer 30–50 hPa above the surface for each stratus **(a)** and stratus-free night **(b)** from −48 to 0 h (color). Trajectories are started at 03:00 UTC. Black solid lines indicate the path of air parcels in IOPs 7 **(a)** and 10 **(b)**, respectively.



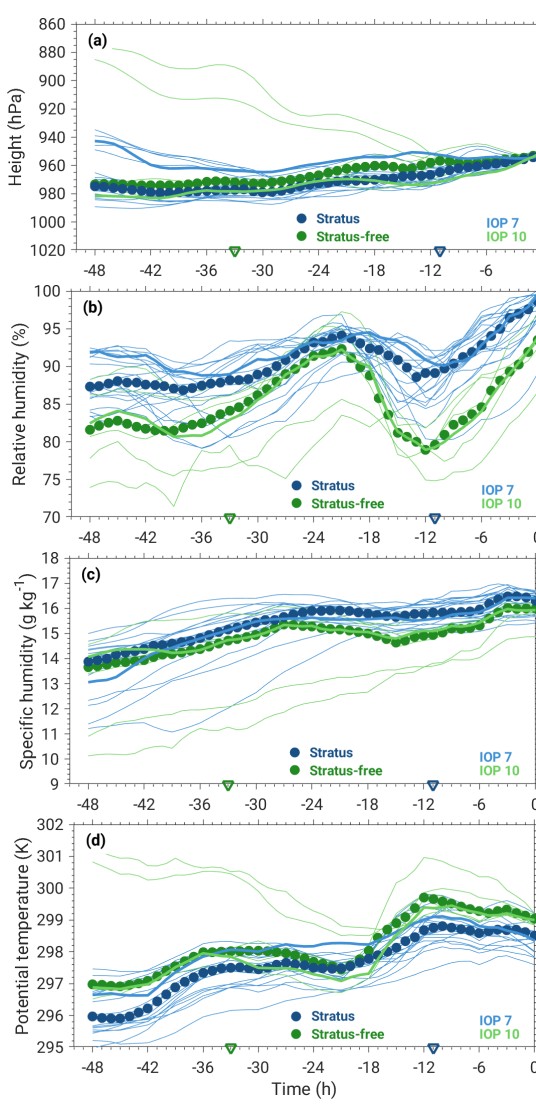

**Figure 13.** Temporal evolution of the pressure **(a)**, relative humidity **(b)**, specific humidity **(c)** and potential temperature **(d)** along the median trajectories averaged in the layer 30–50 hPa above the surface for stratus (blue circles) and stratus-free nights (green circles). The individual trajectories of each stratus and stratus-free night are shown in light blue and green, respectively. The thick blue and green lines indicate trajectories for IOP 7 and 10, respectively, while the blue and green triangles indicate the time when trajectories reach the land for IOP 7 and 10, respectively.