# Peer review of "What controls the formation of nocturnal low-level stratus clouds over southern West Africa during the monsoon season?"

_Atmospheric Chemistry and Physics, 2019_

## Referee Comment (RC1) · Anonymous Referee #1 · 6 Jul 2019

Review of "What controls the formation of nocturnal low-level stratus clouds over southern West Africa during the monsoon season?"

The topic of this paper is the formation (and not formation) of low level stratus clouds at night over West Africa. Low level clouds, not only in this region but in general, are an important research topic as they are one of the largest uncertainties in climate projections. In this paper measurements from a campaign in a region which is otherwise not covered very well by operational measurements have been performed and analyzed. The new data is compared to older observational and modeling studies and improves the understanding of the physical processes. A rather weak point of the manuscript is

the small number of analyzed days. On the one hand, this is understandable, as longer campaigns in remote areas are difficult to carry out and to finance. On the other hand, the results should be seen with caution because the number of cases considered is too small for robust statistics.

This paper should be published after comments listed below have been addressed.

MAJOR COMMENT 1: Backward trajectories have been calculated from ERA5 reanalysis data to investigate how air parcels evolve until the formation of stratus clouds. The method used does not seem to be applicable in the boundary layer or least requires further explanation:

1.1 Data: The data used is not specified well. ERA5 consists out of a high-resolution run and a lower resolution ensemble. Non of these has a resolution of 0.5° as mentioned in the text. That means data has been up- or down-sampled. The question is, when the high-resolution data set has been used, why was it interpolated to a lower resolution (which makes the method even more questionable)? If the low-resolution ensemble has been used, why is there no uncertainty shown? The data is available hourly, why was 3-hourly data used (reducing temporal resolution is not good for trajectories)?

1.2 Does the model IFS, which is used for ERA5, belong to the state-of-the-art models mentioned in the abstract, which have issues representing boundary layer clouds? If so, is it reasonable to believe that small differences in the conditions between stratus and stratus-free are well represented within this model? Reading Page 10, Line 5, I would guess that this is not the case. Related question: have observations from the campaign been sent to ECMWF for assimilation?

1.3 The main problem I see with the method of calculating backward trajectories is really the temporal and spatial resolution of the model. Time-scales in the boundary layer range from minutes to maybe an hour. To resolve that, you would need LES simulations. The content of air parcels you try to track is completely replaced by vertical

turbulent mixing within 3 hours. This process is not resolved by a model at the given resolution. That means, already after the first time step of your backward trajectory, you don't know anymore where your air parcel went and you also don't know where the air parcel you are looking at instead is coming from. Statements like "... situations when air parcels originate above the marine boundary layer ..." (Page 11, Line 4) are just impossible to make as you lose track of your parcel much too fast.

MINOR COMMENTS:

Page 2, Line 31: Typo: "otuput" instead of "output".

Page 4, Line 20: Microwave radiometer are usually not able to distinguish multiple vertical layers. What does high resolution mean in this context?

Page 6, Lines 33-35: How does the presence of LLC keep TKE up? I would expect the reverse connection: Continuous high values of TKE keep LLC forming.

Page 7, Line 12: The term "vortex period" shows up a few times, but the first time it is explained is in the summary.

Page 7, Line 28: With one measurement at Savè, you should not talk about pronounced differences.

Figure 3: The map is hard to read, especially but not only for color-blind people. You could think about using traditional display of octas in meteorological charts.

Figure 4 and 6: How are the 25th and 75th percentiles calculated from six values (six cloud free nights)?

Figure 5: What is the source of the potential temperature? Radiosondes? Microwave?

Figure 9: Is this a composite of multiple stations or is it just Savè?

Data availability: - Does the "KASS-D" data belong to the campaign data? - The web page mentioned for measurements from the campaign offers data sets from a number

of projects, but the page for "DACCIWA" is completely empty.
* * *

---

## Referee Comment (RC2) · Anonymous Referee #2 · 15 Jul 2019

General comments

This discussion paper questions the origin of nocturnal low-level clouds which are found in the boreal summer over southern West Africa. The topic has attracted a lot of publications in the last 12 years, with important advances but still uncertainties on the mechanisms explaining stratus formation. The authors clearly state the pending science questions they seek to answer. The interest of this paper is that it considers a comprehensive observation data set collected during a field campaign in 2016, as well as other relevant data like the recently released ERA5 reanalyses for instance. The authors justify this additional work by the fact that these new data provide observational

verification of modeling studies, and expand on the earlier observational studies based on data collected further north. Their aim is to enable a comprehensive overview for the whole of West Africa, which is partly achieved. There is still some limitation on the full bearing of the results since the analysis relies on the stratus detection at the station of Savè only, but the authors verified that, at least in the central part of the region, there is some spatial coherence in the occurrence of stratus / stratus-free (S/SF) nights.

The paper brings very useful new material which help to document the local vertical structure of the atmosphere on clear and cloudy nights, as well as the associated large-scale dynamics, using distant radiosonde measurements from both coastal and inland stations as well as the reanalysis data. Convincing observations are obtained on the role of the specific humidity of the airmass, and to some extent of the direction of the large-scale wind flow, which demonstrates the part played by synoptic-scale conditions. Results on the NLLJ are a bit more fragile. The timing of the jet onset is suggested to have an effect on cold-air advection, but the evidence is mostly based on case studies during intensive observation periods, and as discussed in section 7 there are some discrepancies with earlier findings. However the authors are generally careful in their well-balanced conclusions. On the whole the paper provides a wealth of new results which help to gain a better understanding of the low level clouds, and which are definitely worth to be published.

Specific comments

1. p.3, l.20 : the authors should perhaps downplay a bit the objectives " provide a comprehensive overview for the whole West Africa'.

2. Most of the analysis is based on the initial identification of stratus and stratus-free nights using ceilometer data. A brief discussion of the accuracy of these data would be useful.

3. The number of stratus-free nights (6) is small. The authors are aware in their conclusions of the limitations attached to this small sample, but an earlier discussion

on the issue would be welcome. For instance, is there any risk of seasonal bias? Several of these nights are located in the early part of the season.

4. The representativeness of the S/SF nights seems to be limited to the region around Benin/Togo, as shown in section 3. Perhaps the samples are a bit small, but it could be useful to indicate whether there is any statistical difference between the cloud covers averaged over S and SF nights at each station in figure 3.

5. p.6 l.10 "an earlier arrival of the maritime inflow": how can we be certain that this flow is of maritime origin, and that the airflow observed earlier is not?

6. Section 5 (IOP cases): it shows very interesting observations, whose interpretation is perhaps more straightforward than for the composite analyses, but care has to exerted on a possible generalization from only two cases. The horizontal advection calculations refer to very different periods of time; is it relevant?

7. p.9, l.32: The pattern shown on the ERA5 wind composite difference between S and SF nights is interesting, with wind anomalies orthogonal to the monsoon flow. Is there any possible connection with the sea breeze?

8. Backward trajectories: how consistent are these findings with the hypothesis of a southern Africa origin of the air mass, as discussed in section 7?

9. Section 7: The authors propose some explanations to the discrepancies found between their study and earlier work. Perhaps the small sample of SF nights may also be taken into account. The authors actually underline this point in their conclusions.

Technical comments

- p.3, section 3.1: we believe that all the data discussed in this section were collected at Savè ?

- p.5, l.31: misprints : processes ; possibly

- section 6 p.9 : it is believed that the ERA5 analyses refer to the composites studied

in sections 3-4, not the IOP periods ?

- backward trajectories : p.5 refers to the levels from 10 to 100 hPa above the surface (i.e., about which elevation?), and p. 10 from 30 and 50 hPa.

- p.10, l.23 : "the two trajectories (during two consecutive nights...)" : unclear what are these trajectories and why they are only two.

- figure 4 caption : supersites

- figure 5: which location ?

- figure 6: (a) is supposed to be wind speed and (b) temperature, not the reverse

- figure 9: it seems that the scale of the vectors is not the same in (a) and (b), if one considers colour shadings. This may not be a major problem, but you need to draw the attention of the reader on it in the caption. We also guess that (c) and (d) display horizontal wind vectors.
* * *

---

## Author Comment (AC1) · 9 Sep 2019

Please see our responses in the attached pdf file.

Please also note the supplement to this comment:
https://www.atmos-chem-phys-discuss.net/acp-2019-537/acp-2019-537-AC1-supplement.pdf

---

## Author Response (AR1)

**Response to the referee comments on "What controls the formation of nocturnal low-level stratus clouds over southern West Africa during the monsoon season?" by K. Babić et al.**

Dear Co-Editor,

We are thankful for the valuable suggestions and comments by the referees. Please find below our point-by-point responses to the referees' comments. The review comments are shown in black, while our responses to individual comments are in blue. In the revised manuscript new or changed text is highlighted as blue.

Sincerely,

Karmen Babić on behalf of all coauthors

**Referee #1:**

Review of "What controls the formation of nocturnal low-level stratus clouds over southern West Africa during the monsoon season?"

The topic of this paper is the formation (and not formation) of low level stratus clouds at night over West Africa. Low level clouds, not only in this region but in general, are an important research topic as they are one of the largest uncertainties in climate projections. In this paper measurements from a campaign in a region which is otherwise not covered very well by operational measurements have been performed and analyzed. The new data is compared to older observational and modeling studies and improves the understanding of the physical processes. A rather weak point of the manuscript is the small number of analyzed days. On the one hand, this is understandable, as longer campaigns in remote areas are difficult to carry out and to finance. On the other hand, the results should be seen with caution because the number of cases considered is too small for robust statistics. This paper should be published after comments listed below have been addressed.

We would like to thank Referee #1 for the time and effort spent in reviewing our paper, for the thorough comments and for the useful suggestions. The Reviewer's acknowlegment on the importance of the topic of this paper is highly appreciated. We have produced a revised paper based on the comments and suggestions, while detailed response to each comment is provided below.

**Major Comment:**

1. Backward trajectories have been calculated from ERA5 reanalysis data to investigate how air parcels evolve until the formation of stratus clouds. The method used does not seem to be applicable in the boundary layer or least requires further explanation:

   1.1. Data: The data used is not specified well. ERA5 consists out of a high-resolution run and a lower resolution ensemble. Non of these has a resolution of 0.5° as mentioned in the text. That means data has been up- or down-sampled. The question is, when the high-resolution data set has been used, why was it interpolated to a lower resolution (which makes the method even more questionable)? If the low-resolution ensemble

has been used, why is there no uncertainty shown? The data is available hourly, why was 3-hourly data used (reducing temporal resolution is not good for trajectories)? In the revised paper we have calculated new backward trajectories using the highest possible spatial and temporal resolutions of the ERA5 data set, i.e. 30 km and 1 hourly analysis fields. In Section 2.2 we have modified the description of the ERA5 data accordingly. Also, Figs. 11, 12 and 13 have been modified as well.

1.2. Does the model IFS, which is used for ERA5, belong to the state-of-the-art models mentioned in the abstract, which have issues representing boundary layer clouds? If so, is it reasonable to believe that small differences in the conditions between stratus and stratus-free are well represented within this model? Reading Page 10, Line 5, I would guess that this is not the case. Related question: have observations from the campaign been sent to ECMWF for assimilation?
During the DACCIWA field campaign, tailored forecasts were made using different operational NWP models, namely, ICON, COSMO, IFS and Met Office Unified Model. Kniffka et al. (submitted) provides a comprehensive evaluation of the NWP models using radiosondes, ground-based measurements and satellite data. Their results showed that low-level clouds (LLC) are generally underestimated in the models, which leads to an excess solar radiation at the earth's surface. During the DACCIWA campaign, LLC cover was underestimated by on average 11% for all models. They also show, based on the statistical day-to-day grid point based analysis, that the models perform best in forecasting temperature, with the smallest spread between the models. In our analysis, composite analysis of ERA5 data shows a good agreement with observed specific humidity values. On the other hand, comparing ERA5 temperature fields with our point observations at radiosonde stations suggest that the differences between stratus and stratus-free nights in ERA5 are of the opposite sign, suggesting warmer stratus-free nights. The absolute magnitude of the difference is between 0.5 K at stations closer to the coast and up to 1 K at Savè supersite and Parakou. However, we note that these differences are on the order of the instrument measurement accuracy. Additionally, we assume that the agreement would have been better if we could compare ERA5 data with more observations across the larger region. Based on these considerations, we believe that the models can capture the differences in conditions during stratus and stratus-free nights. As far as we know, unfortunately neither radiosonde nor surface observations from the DACCIWA campaign have not been assimilated in ERA5 data set.

1.3. The main problem I see with the method of calculating backward trajectories is really the temporal and spatial resolution of the model. Time-scales in the boundary layer range from minutes to maybe an hour. To resolve that, you would need LES simulations. The content of air parcels you try to track is completely replaced by vertical turbulent mixing within 3 hours. This process is not resolved by a model at the given turbulent mixing within 3 hours. This process is not resolved by a model at the given resolution. That means, already after the first time step of your backward

trajectory, you don't know anymore where your air parcel went and you also don't know where the air parcel you are looking at instead is coming from. Statements like ... situations when air parcels originate above the marine boundary layer ... (Page 11, Line 4) are just impossible to make as you lose track of your parcel much too fast. We agree with the Reviewer that in order to track air parcels in the atmospheric boundary layer one needs to use an LES output with high spatial and temporal resolutions, i.e. on the order of tens of meters and minutes. However, since these simulations are not available and also would be extremely costly to produce them, in the revised manuscript we use ERA5 data with 30 km spatial resolution and 137 vertical levels and 1 hourly temporal resolution. Our intention is not to track individual air parcels, but to identify the pathways of air masses which reach Savè supersite, in order to investigate how synoptic disturbances/large-scale flow over south West Africa affect the boundary layer conditions. In the revised paper, we have calculated new backward trajectories using the highest possible spatial and temporal resolution from ERA5, and in Section 2.2 we have introduced a discussion on the limitations of this method when it comes to tracking individual air parcels in the boundary layer. Figures 11, 12 and 13 have been updated, accordingly.

**Minor Comments:**

Page 2, Line 31: Typo: otuput instead of output.

Corrected.

Page 4, Line 20: Microwave radiometer are usually not able to distinguish multiple vertical layers. What does high resolution mean in this context?

The microwave radiometer (HATPRO) provided temperature profiles with enhanced accuracy using low-elevation boundary-layer scans, which allows to resolve the ABL, with a temporal resolution of 15-min (Dione et al. 2019). Thus, the HATPRO measurements provided the information on the thermal structure of the boundary layer at high temporal resolution, which is important in order to capture the arrival of the Gulf of Guine maritime inflow. This information is introduced in Section 2.1 of the revised paper.

Page 6, Lines 33-35: How does the presence of LLC keep TKE up? I would expect the reverse connection: Continuous high values of TKE keep LLC forming.

In Section 4.1 we compare conditions between stratus and stratus-free nights and we do not discuss on the factors causing the formation and maintenance of stratus clouds. Certainly there exists a positive feedback between the high values of TKE and LLC, and in previous work (cf Adler et al. 2019) we have found that higher values of TKE are correlated with lower values of the bulk Richardson number, which typically occurs in situations with coupled sub-cloud layer. Comparison of stratus and stratus-free nights shows increased values of the TKE related to the onset of NLLJ, however, in stratus-free nights the TKE decreases in the second part of the night due to the stronger surface cooling, leading to the build-up of the near-surface stability, which dampens vertical motions and turbulent mixing. On the other hand, the LLC reduce the cooling of the surface, resulting in the near-neutral near-surface layer, which helps to keep increased mixing and TKE values.

Page 7, Line 12: The term vortex period shows up a few times, but the first time it is explained is in the summary.

The explanation of the vortex phase is now introduced already in the Introduction section.

Page 7, Line 28: With one measurement at Savè, you should not talk about pronounced differences.

This sentence is now rephrased to indicate that the differences at Savè and Parakou are slightly larger than at the coastal stations.

Figure 3: The map is hard to read, especially but not only for color-blind people. You could think about using traditional display of octas in meteorological charts.

We have followed Reviewer's advice and created a new Figure 3 using a color map which is suitable for people with color vision deficiency (Stauffer et al. 2015). The color map is created using a palette creator at `http://hclwizard.org/` web site. We have also removed the information on topography, since it is available in Fig. 1, in order to increase the readability.

Figure 4 and 6: How are the 25th and 75th percentiles calculated from six values (six cloud free nights)?

The percentiles for six cloud-free nights are calculated in the way that values are first sorted and 25th percentile shows the values of the second profile and 75th percentile is the fifth profile. We are aware that it is not completely appropriate to use percentiles when having a small number of profiles, however, we wanted to avoid using the mean and standard deviation, since these might be affected by one extreme value. On the other hand, we wanted to show the degree of variability in our data set, but showing all individual profiles would result in a very busy figure, which we wanted to avoid.

Figure 5: What is the source of the potential temperature? Radiosondes? Microwave?

The potential temperature shown in Fig. 5 is from microwave radiometer measurements. This information is introduced in the revised version of the paper.

Figure 9: Is this a composite of multiple stations or is it just Savè?

Figure 9 shows measurements at Savè supersite during two different IOPs. This information is now included in the caption.

Data availability:

- Does the "KASS-D" data belong to the campaign data?

  The data from the KASS-D data base were not collected as part of the DACCIWA campaign. The KASS-D data base is a result of long-standing collaborations with African national weather services and African researchers. This data base is not publicly available. It is maintained by researchers at KIT and anyone interested in the data should contact Prof. Andreas Fink. This information is added in the *Data availability* part.

- The webpage mentioned for measurements from the campaign offers data sets from a number of projects, but the page for "DACCIWA" is completely empty.

  The data sets from the DACCIWA campaign are freely available to everyone who has registered at the BAOBAB web site. DACCIWA licence is CC-By for all data sets, with authentication access, meaning all data sets are public and there is no restriction on registration. The required registration in order to use the DACCIWA data set makes it

possible to make the statistics on the data usage and keep contact with users.

**Referee #2:**

**General comments:**

This discussion paper questions the origin of nocturnal low-level clouds which are found in the boreal summer over southern West Africa. The topic has attracted a lot of publications in the last 12 years, with important advances but still uncertainties on the mechanisms explaining stratus formation. The authors clearly state the pending science questions they seek to answer. The interest of this paper is that it considers a comprehensive observation data set collected during a field campaign in 2016, as well as other relevant data like the recently released ERA5 reanalyses for instance. The authors justify this additional work by the fact that these new data provide observational verification of modeling studies, and expand on the earlier observational studies based on data collected further north. Their aim is to enable a comprehensive overview for the whole of West Africa, which is partly achieved. There is still some limitation on the full bearing of the results since the analysis relies on the stratus detection at the station of Savè only, but the authors verified that, at least in the central part of the region, there is some spatial coherence in the occurrence of stratus / stratus-free (S/SF) nights. The paper brings very useful new material which help to document the local vertical structure of the atmosphere on clear and cloudy nights, as well as the associated large-scale dynamics, using distant radiosonde measurements from both coastal and inland stations as well as the reanalysis data. Convincing observations are obtained on the role of the specific humidity of the airmass, and to some extent of the direction of the large-scale wind flow, which demonstrates the part played by synoptic-scale conditions. Results on the NLLJ are a bit more fragile. The timing of the jet onset is suggested to have an effect on cold-air advection, but the evidence is mostly based on case studies during intensive observation periods, and as discussed in section 7 there are some discrepancies with earlier findings. However the authors are generally careful in their well-balanced conclusions. On the whole the paper provides a wealth of new results which help to gain a better understanding of the low level clouds, and which are definitely worth to be published.

We would like to thank Referee #2 for the time and effort spent in reviewing our paper, for the thorough comments and for the useful suggestions. We appreciate the Reviewer's acknowlegment on the role of our results in gaining better understanding of LLC. We would like to mention that the results regarding the impact of the nocturnal low-level jet (NLLJ) timing on the cold-air advection are supposed to ilustrate how deviation from the typical conditions impacts the formation of LLC. Namely, in this case the onset of NLLJ was unusually late compared to the average arrival time (around 20:00 UTC, according to Fig. 5). The role of NLLJ on the formation of stratus clouds was investigated in detail in previous work, e.g. in Adler et al. (2019); Babić et al. (2019); Dione et al. (2019). We have produced a revised paper based on the comments and suggestions, while detailed response to each comment is provided below.

**Specific comments:**

1. p.3, l.20 : the authors should perhaps downplay a bit the objectives "provide a comprehensive overview for the whole West Africa".
   The objective has been reformulated to provide the overview for the southern West Africa.

2. Most of the analysis is based on the initial identification of stratus and stratus-free nights using ceilometer data. A brief discussion of the accuracy of these data would be useful.
   More information on the accuracy of the ceilometer measurements is introduced in Section 2.1 in the revised paper. Please note, the identification of stratus and stratus-free nights using ceilometer data is quite straightforward and is not very sensitive to the accuracy of the ceilometer measurements. The criteria we used to select the stratus nights based on ceilometer data is that the cloud base height (CBH) is below 500 m a.g.l., that the stratus deck is persistent and continuous for the largest part of the night. An example of ceilometer backscatter and derived CBH for stratus case (IOP 7) and stratus-free night (IOP 10) is shown in Fig. 1.

[Figure]

**Figure 1:** Time series of ceilometer backscatter (color) and CBH (black dots) derived from the backscatter profiles.

3. The number of stratus-free nights (6) is small. The authors are aware in their conclusions of the limitations attached to this small sample, but an earlier discussion on the issue would be welcome. For instance, is there any risk of seasonal bias? Several of these nights are located in the early part of the season.
   One of six stratus-free nights occurred in mid-June (14-15 June), which according to Knippertz et al. (2017) is before the post-onset monsoon phase. In our analysis, only the near-surface measurements at Savè include this one night from pre-monsoon period, while radiosonde and remote sensing measurements started after 18 June. However, considering that the DACCIWA campaign did not cover the whole monsoon season, it is difficult to

say whether there is a seasonal bias. In previous studies, which analyzed longer periods and could identify larger number of clear nights, e.g. Schrage et al. (2007) found that the differences between cloudy and clear nights are most likely related to day-to-day synoptic changes in the monsoon structure. Therefore, it is unlikely that our results are influenced by the seasonal bias. We follow the Reviewer's advice and introduce the discussion on the limitations related to the small cloud-free sample already in Section 3.

4. The representativeness of the S/SF nights seems to be limited to the region around Benin/Togo, as shown in section 3. Perhaps the samples are a bit small, but it could be useful to indicate whether there is any statistical difference between the cloud covers averaged over S and SF nights at each station in figure 3.

   We have followed Reviewer's suggestion and tested the difference in cloud cover reported at synop stations for stratus and stratus-free nights using the two-sample $t$-test. The results suggest that at 80 % of the stations (43 out of 53) accross the area the differences are significant at 95 % level. This test indicates that at 10 stations the difference is not significant, out of which 7 is in Ghana. Moreover, the test suggest that synop observations at Savè are not significantly different. Considering that the selection of stratus and stratus-free nights was done based on the reliable ceilometer measurements at Savè supersite, we doubt the applicability of such statistical test for these observations. Observations of cloud base height and cover at synop stations are probably less accurate than ceilometer, cloud camera and radiation data available at Savè. Nevertheless, the synop data are the only one which provide information about the spatial distribution of LLC (since the satellite data could not be used due to present mid- and high clouds during stratus-free nights) and, therfore, we think they are valuable to use in this study. Thus, Fig. 3 is presented in order to ilustrate cloud cover in a qualitative sense over larger area. The two-sample $t$-test indicates that differences for stations at the coast and above 8° N are significant.

5. p.6 l.10 an earlier arrival of the maritime inflow: how can we be certain that this flow is of maritime origin, and that the airflow observed earlier is not?

   The main characteristic of the maritime inflow arrival is the decrease of temperature, since it describes the front which separates the cool maritime air mass and the warmer air mass over the land and is located several tens of kilometers from the coast in the late afternoon (Adler et al. 2017; Deetz et al. 2018). The cool maritime air is transported farther inland with the southwesterly monsoon flow during the late afternoon and evening. Adler et al. (2019) found for IOP nights at Savè supersite a concurrent increase in wind speed (with the profile showing a NLLJ structure) and a decrease in temperature in a several hundred meters deep layer during the first half of the night. The changes in atmospheric conditions related to the maritime inflow arrival for one case study are illustrated in Babić et al. (2019). Therefore, the sentence on p. 6, line 10 is reformulated and the discussion of the earlier maritime inflow arrival for stratus-free nights is introduced after the discussion of the temperature change for stratus-free nights (in the revised paper pg. 6, line 33 and pg. 7, lines 1-2).

6. Section 5 (IOP cases): it shows very interesting observations, whose interpretation is perhaps more straightforward than for the composite analyses, but care has to exerted on a possible generalization from only two cases. The horizontal advection calculations refer to very different periods of time; is it relevant?

The calculation of horizontal cold-air advection is done for two different time periods in order to fulfill the underlying assumptions. Namely, in Adler et al. (2019) several assumptions are made: (i) the temperature distribution is homogeneous along the coast and that the zonal temperature gradient and wind component are small; (ii) there is a linear increase in temperature in the south-north direction in the maritime inflow and a constant temperature in the CBL over land; (iii) the maximum inland propagation of the maritime inflow in the late afternoon is marked as the distance where temperature starts to gradually decrease towards the coast; (iv) the maritime inflow propagates with the maximum southerly wind component of the averaged vertical profile, i.e. average of the meridional component of the coastal wind profile in the afternoon and of the wind profile at Savè after the maritime inflow arrives. Therefore, considering that the onset of the NLLJ related to the maritime inflow was after 23:00 UTC for stratus-free case study (IOP 10), different periods were used for the calculation of horizontal cold-air advection. In the revised paper, we introduced the information why horizontal advection is calculated for two different periods (pg. 9).

7. p.9, l.32: The pattern shown on the ERA5 wind composite difference between S and SF nights is interesting, with wind anomalies orthogonal to the monsoon flow. Is there any possible connection with the sea breeze?

The orthogonal wind anomalies over the Gulf of Guinea indicate that the u-wind component is larger for stratus-free nights, thus indicating stronger westerly flow component. On the other hand, the location of the see breeze front is controlled by the v (southerly) wind component, and it depends on the strength of the monsoon flow.

8. Backward trajectories: how consistent are these findings with the hypothesis of a southern Africa origin of the air mass, as discussed in section 7?

The hypothesis about the transport of dry air masses from central/south Africa is tested by additional calculation of backward trajectories. We have used ERA5 data with 30 km horizontal resolution, 137 vertical levels and hourly output to calculate 10-day backward trajectories, which were started on 13 July 2016 at 00:00 UTC, i.e. during the vortex period. The trajectories are started at 10 hPa increments starting at 10 hPa above the surface up to 100 hPa above the surface. The results are shown in Fig. 2. These calculations confirm the validity of the hypothesis: the central/southern African air masses being transported westward, descending from 750 to 850-900 hPa level, and after being entrained in the marine boundary layer are transported farther toward the southern West Africa with southwesterly monsoon flow. This is especially visible for trajectories started from 40 up to 90 hPa above the surface. In Section 7 of the revised paper, we have added the information on the 10-day backward trajectories results from ERA5.

[Figure]

**Figure 2:** Left: 10-day backward trajectories started on 13 July 2016 at 00:00 UTC at different levels (40-100 hPa) above the surface. Right: The time series of pressure and specific humidity along the trajectories at different levels above the surface.

9. Section 7: The authors propose some explanations to the discrepancies found between their study and earlier work. Perhaps the small sample of SF nights may also be taken into account. The authors actually underline this point in their conclusions.

   In the revised paper (in Section 7) we have included the discussion on the small number of observed stratus-free nights as a possible reason for the discrepancies between our results and results from previous studies.

**Technical comments::**

- p.3, section 3.1: we believe that all the data discussed in this section were collected at Savè ?

  Correct. We now introduced the information that this section describes different measurements from Savè supersite.

- p.5, l.31: misprints : processes ; possibly

  Corrected.

- section 6 p.9 : it is believed that the ERA5 analyses refer to the composites studied in sections 3-4, not the IOP periods ?

  Correct. This is made clear by indicating the total number of analyzed cases.

- backward trajectories : p.5 refers to the levels from 10 to 100 hPa above the surface (i.e., about which elevation?), and p. 10 from 30 and 50 hPa.

  In the revised manuscript we have added the explanation for the reason to calculate backward trajectories from 10 to 100 hPa above the surface, as well as for averaging them in the layer between 30 and 50 hPa above the surface, since this layer corresponds to the average stratus layer during the cloudy nights.

- p.10, l.23 : the two trajectories (during two consecutive nights...) : unclear what are these trajectories and why they are only two.
  This sentence is reformulated in the revised paper (please see page 11).

- figure 4 caption : supersites
  Corrected.

- figure 5: which location ?
  The location information is added, it is Savè supersite.

- figure 6: (a) is supposed to be wind speed and (b) temperature, not the reverse
  Corrected.

- figure 9: it seems that the scale of the vectors is not the same in (a) and (b), if one considers colour shadings. This may not be a major problem, but you need to draw the attention of the reader on it in the caption. We also guess that (c) and (d) display horizontal wind vectors.
  Thank you for this remark. We have added the information about the different size of the horizontal wind vectors (arrows) for different cases.

**References**

Adler, B., Kalthoff, N., and Gantner, L.: Nocturnal low-level clouds over southern West Africa analysed using high-resolution simulations, Atmos. Chem. Phys., 17, 899–910, https://doi.org/10.5194/acp-17-899-2017, 2017.

Adler, B., Babić, K., Kalthoff, K., Lohou, F., Lothon, M., Dione, C., Pedruzo-Bagazgoitia, X., and Andersen, H.: Nocturnal low-level clouds in the atmospheric boundary layer over southern West Africa: an observation-based analysis of conditions and processes, Atmos. Chem. Phys., 19, 663–681, https://doi.org/10.5194/acp-2018-775, 2019.

Babić, K., Adler, B., Kalthoff, N., Andersen, H., Dione, C., Lohou, F., Lothon, M., and Pedruzo-Bagazgoitia, X.: The observed diurnal cycle of low-level stratus clouds over southern West Africa: a case study, Atmos. Chem. Phys., 19, 1281–1299, https://doi.org/10.5194/acp-19-1281-2019, 2019.

Deetz, K., Vogel, H., Knippertz, P., Adler, B., Taylor, J., Coe, H., Bower, K., Haslett, S., Flynn, M., Dorsey, J., Crawford, I., Kottmeier, C., and Vogel, B.: Numerical simulations of aerosol radiative effects and their impact on clouds and atmospheric dynamics over southern West Africa, Atmos. Chem. Phys., 18, 9767–9788, https://doi.org/10.5194/acp-18-9767-2018, 2018.

Dione, C., Lohou, F., Lothon, M., Adler, B., Babić, K., Kalthoff, N., Pedruzo-Bagazgoitia, X., Bezombes, Y., and Gabella, O.: Low-level stratiform clouds and dynamical features observed within the southern West African monsoon, Atmos. Chem. Phys., 19, 8979–8997, https://doi.org/10.5194/acp-19-8979-2019, 2019.

Kniffka, A., Knippertz, P., Fink, A. H., Benedetti, A., Brooks, M., Hill, P., Maranan, M., Pante, G., and Vogel, B.: An evaluation of operational and research weather forecasts for southern West Africa using obesrvations from the DACCIWA field campaign in June-July 2016, Q. J. Roy. Meteor. Soc., submitted.

Knippertz, P., Fink, A. H., Deroubaix, A., Morris, E., Tocquer, F., Evans, M. J., Flamant, C., Gaetani, M., Lavaysse, C., Mari, C., Marsham, J. H., Meynadier, R., Affo-Dogo, A., Bahaga, T., Brosse, F., Deetz, K., Guebsi, R., Latifou, I., Maranan, M., Rosenberg, P. D., and Schlueter, A.: A meteorological and chemical overview of the DACCIWA field campaign in West Africa in June–July 2016, Atmos. Chem. Phys., 17, 10 893–10 918, https://doi.org/10.5194/acp-17-10893-2017, 2017.

Schrage, J. M., Augustyn, S., and Fink, A. H.: Nocturnal stratiform cloudiness during the West African monsoon, Meteorol. Atmos. Phys., 95, 73–86, https://doi.org/10.1007/s00703-006-0194-7, 2007.

Stauffer, R., Mayr, G. J., Dabernig, M., and Zeileis, A.: Somewhere Over the Rainbow: How to Make Effective Use of Colors in Meteorological Visualizations, B. Am. Meteorol. Soc., 96, 203–216, https://doi.org/10.1175/BAMS-D-13-00155.1, 2015.